# Scaling Transformers for Low-Bitrate High-Quality Speech Coding

**Julian D. Parker**[*]  **Anton Smirnov**  **Jordi Pons**  **CJ Carr**  **Zack Zukowski**
**Zach Evans**  **Xubo Liu**[*]
Stability AI
{julian.parker, xubo.liu}@stability.ai

## Abstract

The tokenization of speech with neural audio codec models is a vital part of modern AI pipelines for the generation or understanding of speech, alone or in a multimodal context. Traditionally such tokenization models have concentrated on low parameter-count architectures using only components with strong inductive biases. In this work we show that by scaling a transformer architecture with large parameter count to this problem, and applying a flexible Finite Scalar Quantization (FSQ) based bottleneck, it is possible to reach state-of-the-art speech quality at extremely low bit-rates of 400 or 700 bits-per-second. The trained models strongly out-perform existing baselines in both objective and subjective tests.

## 1 Introduction

Compressed coding of audio and speech data in digital format has been an active area of research since the 1970s , and reached particular prominence in the late 1990s with the emergence of mp3 (Painter & Spanias, 2000). Research into improving the sound quality and compression ratio of such codecs (mainly using signal processing techniques) has continued (Valin et al., 2016). The main purpose of these codecs is to improve the efficiency of transmission and storage of what is traditionally a data-intensive medium.

In recent times, the research community began to apply the techniques of machine learning to the audio coding problem (Zeghidour et al., 2021). These models are referred to as *neural audio codecs* (NACs). Initially the goal of these models was similar to traditional audio codecs, which aim to maximize compression and audio quality at low computational cost. However, a paradigm shift occurred with the proposal of powerful generative models utilizing the token sequences produced by these codecs (Borsos et al., 2023a; Wang et al., 2023; Borsos et al., 2023b). With the arrival of these models and the plethora of new use-cases they encompass, the design goals of NACs have shifted to be less concerned with computational complexity, and more concerned with pushing compression (especially in the temporal dimension) to the maximum level possible.

Our goal is to design a speech codec model in the spirit of this paradigm shift, whose primary purpose is to be used in combination with modern generative architectures for generation or understanding of speech signals. We make the observation that in a typical modern generative pipeline for speech there may be models totalling billions of parameters, a tiny fraction of which is usually dedicated to the codec model. There is therefore some headroom to increase the size of this component without overly impacting overall computational burden. This opens up scaling of the codec model size as a route to higher quality audio and higher compression levels.

Neural audio codec models have largely been based on convolutional or recurrent architectures, which can be challenging to scale to larger model sizes without placing restrictions on the architecture. Even with such restrictions, the largest successful purely convolutional networks are generally below 1B parameters (Woo et al., 2023). Transformers (Vaswani, 2017) have shown the ability to scale to billions of parameters in many domains (Hoffmann et al., 2022), but have not been fully utilized in a codec context yet. Recent work has also deployed transformer blocks in the bottleneck of a convolutional codec, showing improvements in compression ratio (Défossez et al., 2024).

---

[*]Equal contribution

However, transformers have not so far been deployed as the main component of a codec model. One major contribution of this work is to design a new codec architecture that is predominantly transformer-based, and scale such an architecture into the 1B parameter range.

The majority of current codecs utilize a Residual Vector Quantizer (RVQ) (Zeghidour et al., 2021) in some form. This is effective in maximizing the expressivity of the bottleneck for a given bit-rate, but presents a number of challenges for generative modeling. One challenge is that it produces many parallel hierarchical streams of tokens. The causal relationship between the streams introduces a variety of complications that must be accounted for during training and inference (Borsos et al., 2023a; Copet et al., 2023; Défossez et al., 2024). An additional challenge is that VQs and RVQs can suffer from poor or inconsistent codebook utilization, making the process of learning the token distribution more difficult and prone to bias. In this work we address some of the issues of VQ and RVQ by instead adopting a quantization scheme derived from Finite Scalar Quantization (FSQ) (Mentzer et al., 2023), and a novel post-hoc method decomposing FSQ into low-order residuals.

We demonstrate how these contributions enable the training of a waveform codec model that achieves high compression for speech, with ultra-low bitrates of 400 bps and 700 bps, while still preserving good audio quality.

Code and models will be released at: `github.com/Stability-AI/stable-codec`.

## 2 RELATED WORK

### 2.1 NEURAL AUDIO CODECS

The dominant paradigm for training NACs has so far been based on the VQ-VAE structure, consisting of a classic autoencoder-like structure of encoder and decoder model with an information bottleneck placed in between them in the form of a quantizer. Soundstream (Zeghidour et al., 2021) was the first example of such a model aimed at handling varying bit-rates and types of audio with a single model. Soundstream introduced an adversarial loss in addition to reconstruction loss, and residual vector quantization (RVQ) for use in the bottleneck. EnCodec (Défossez et al., 2022) proposed a number of improvements to this formulation and achieved higher audio quality. SpeechTokenizer (Zhang et al., 2023b), building on Encodec, introduces the use of semantic tokens in the first channel of discrete RVQ codecs, bridging the gap between text tokens and acoustic tokens for speech coding.

DAC (also known as improved RVQGAN) (Kumar et al., 2023) investigated several design choices in this type of NAC, including the introduction of periodic inductive biases and improvements in codebook utilization. This approach achieved notable performance, compressing 44.1 kHz audio into discrete codes at an 8 kbps bitrate. While DAC delivers high-quality reconstruction at this compression level, its bitrate remains relatively high for generative audio modeling, requiring over 700 tokens per second for 44.1 kHz audio due to the large number of residual tokens.

### 2.2 LOW BITE-RATE SPEECH CODING

Recently, there has been growing interest (Li et al., 2024; Liu et al., 2024a; Défossez et al., 2024) in optimizing bitrate efficiency in NACs while maintaining high reconstruction quality. Such low-bitrate, high-fidelity codecs are particularly crucial for improving efficiency and reducing latency in generative audio modeling. However, achieving extremely low bitrates (such as below 1 kbps for 16 kHz audio) remains challenging due to the complexities involved in accurately compressing high-frequency components in the audio waveform.

SingleCodec (Li et al., 2024) addressed neural speech coding by proposing an enhanced VQ-VAE combined with bidirectional LSTM for mel-spectrogram compression, achieving a notably low bandwidth of 304 bps for 24 kHz speech mel-spectrogram coding, followed by BigVGAN (Lee et al., 2022) as a vocoder for waveform reconstruction. Inspired by recent advances in generative models, SemantiCodec (Liu et al., 2024a) offers a different approach by leveraging a latent diffusion model to generate latent features from a pre-trained mel-spectrogram VAE (which also requires a vocoder for waveform reconstruction). The diffusion model is conditioned on k-means clustered audio tokens derived from a pre-trained AudioMAE encoder. SemantiCodec supports low bitrates

ranging from 0.31 kbps to 1.43 kbps for 16 kHz speech mel-spectrogram coding, offering a promising solution for maintaining high reconstruction quality at extremely low bitrates.

Mimi (Défossez et al., 2024) is a recent end-to-end waveform codec for speech based on Sound-Stream and Encodec. Mimi introduces transformer layers around the RVQ bottleneck between the convolutional encoder and decoder. By scaling its training data to 7 million hours, Mimi has achieved impressive performance in neural speech coding, operating at 1.1 kbps with a 12.5 kHz latent for 24 kHz speech in a causal way, utilizing 8 tokens per latent frame (100 tokens per second).

### 2.3 GENERATIVE MODELS FOR AUDIO AND SPEECH

Autoregressive models can operate directly on quantized audio waveforms, but can be slow during inference (Oord et al., 2016). Recent models, such as VALL-E (Wang et al., 2023), AudioLM (Borsos et al., 2023a), MusicGen (Copet et al., 2023), and VQ-VAE-based approaches for sound synthesis (Liu et al., 2021), improve efficiency by instead modeling quantized latent sequences. Non-autoregressive models (Oord et al., 2018) and adversarial audio synthesis (Donahue et al., 2018) were developed to overcome the inefficiencies of autoregressive models. Recent non-autoregressive models such as VampNet (Garcia et al., 2023), SoundStorm (Borsos et al., 2023b), or StemGen (Parker et al., 2024) are based on masked token modeling (Chang et al., 2022). End-to-end diffusion modeling can also be computationally demanding (Rouard & Hadjeres, 2021; Pascual et al., 2023). Recent efficiency improvements rely on latent diffusion models (Liu et al., 2023; 2024b; Yuan et al., 2024; Evans et al., 2024a;b;c; Yang et al., 2024), which often rely on VAEs for latent encoding. The recent growth of multi-modal and speech-first generative models such as SpeechGPT (Zhang et al., 2023a), LLaMA3 (Dubey et al., 2024) and Moshi (Défossez et al., 2024) is also heavily reliant on tokenized representations of speech and audio. As such, learning quantized or continuous latent spaces with codecs is crucial for advancing audio and speech generation.

## 3 ARCHITECTURE

The architecture of the codec is shown in overview form in Fig. 1. We will discuss the design of the encoder and decoder sections with FSQ-based bottleneck separately.

### 3.1 ENCODER AND DECODER

Our encoder and decoder structures are designed to look very similar to a standard transformer architecture. Both consist of multiple blocks, each operating at a specific temporal resolution. These sections consist of a strided 1d dense convolution layer (for downsampling in the encoder) or its transposed equivalent (for upsampling in the decoder) and a chain of relatively standard transformer blocks. The only difference between the encoder and decoder architecture is that the downsampling or upsampling layer is placed in a different location—in the encoder at the start of the block, and in the decoder at the end of the block. This maintains symmetry of the architecture. The stacked transformer blocks consist of a self-attention section and a feedforward section, with pre-norm placement of layer norm blocks. The layer norm blocks are configured with a higher than standard $\epsilon$ as discussed in Appendix B.1. In addition, the self-attention utilizes QK-norm. The feedforward block consists of a reverse bottleneck with a gated MLP, utilizing the SiLU activation function. Both attention blocks and feedforward blocks are followed by LayerScale (Touvron et al., 2021), to further stabilize training. The self-attention uses a sliding window to restrict receptive field and aid generalization of the architecture to arbitrary length sequences. The self-attention mechanism incorporates Rotary Positional Embeddings (RoPE) (Su et al., 2024) and operates without a causal attention mask. However, a causal variant suited for streaming purposes is possible with relatively minor modifications, as described in Appendix A.4. We further examine the model's receptive field, causality, and latency in Appendix B.2.

In contrast to convolutional architectures, we want the majority of temporal downsampling or upsampling of the signal to occur at the input or output to the architecture. This is to avoid feeding very small dimension embeddings to the transformer blocks, and also to limit sequence length. Only minimal further resampling happens within the architecture using the strided convolutions and transposed convolutions in each encoder or decoder block. To achieve this we can use any filter-bank representation of the input signal which conforms to perfect reconstruction criteria. The

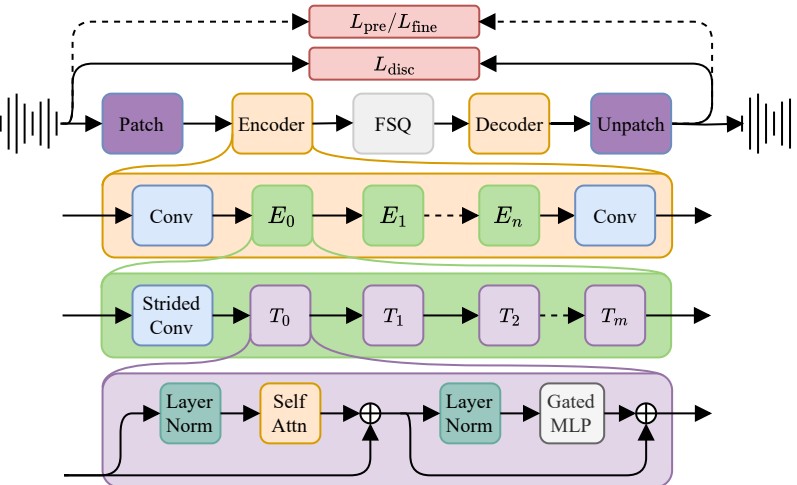

Figure 1: Architecture of the proposed model. Detail is shown for the encoder block and sub-blocks. The decoder block is configured identically to the encoder block, with the exception of the strided convolution, which is replaced with its transposed equivalent and moved to the end of the $T_m$ blocks.

details of this choice are discussed in Appendix B.4. Following the conclusions of this analysis and taking inspiration from Vision Transformer (ViT) architectures (Dosovitskiy et al., 2021), we utilize sequence-wise patching of the signal before passing to the encoder.

Additionally we utilize dense 1d convolutional blocks at the inputs and outputs of the encoder and decoder structure. These blocks map between the embedding dimension used within the transformer (which is uniform) and the required dimension for the input/output patches and the latent representation used in the bottleneck. All convolutional layers use a weight-normalized parameterization.

We call the resulting architecture a Transformer Audio AutoEncoder (TAAE).

A major distinction between TAAE and traditional CNN-based codecs is the extensive use of transformer layers in TAAE, which results in a larger model size compared to CNN-based codecs Zeghidour et al. (2021); Défossez et al. (2022); Kumar et al. (2023). CNN-based models leverage convolutional operations, which offer a strong inductive bias and high parameter efficiency. In contrast, the TAAE uses a transformer-based architecture, providing enhanced scalability, albeit with reduced parameter efficiency. An explanation of these differences and discussion comparing convolution and attention mechanisms can be found in the App B.3.

## 3.2 DISCRETE BOTTLENECK

In order to mitigate the inherent problems of VQ and RVQ quantization, we employ a modified version of Finite Scalar Quantization (FSQ) (Mentzer et al., 2023). Instead of a learnable codebook of embeddings connected to particular tokens as in VQ/RVQ, FSQ derives a token sequence by projecting the latent representation to a low-dimensional space, then scalar quantizing each dimension of this space in regular intervals. Each combination of quantized levels can then be mapped to a unique integer value, producing the tokenization. FSQ is known to exhibit almost full codebook utilisation even with very large codebook sizes (e.g., $2^{18}$) (Mentzer et al., 2023).

We make some modifications to the FSQ algorithm to preserve symmetry of the quantized latents around the origin for any number of levels. Our formulation for the scalar quantizer function $Q_L$ for a given fixed number of levels $L$, applied to some scalar $x$ is given by:

$$Q_L(x) = \frac{2}{L-1} \lfloor (L-1)\frac{\tanh x + 1}{2} + \frac{1}{2} \rfloor - 1 \qquad (1)$$

This scalar quantization function is applied (potentially with different L per dimension), to the elements of a latent vector $\mathbf{z}$ to produce the quantized latent.

To train with this scalar quantizer, we use a hybrid approach. Some percentage of the time we emulate the effect of quantization by adding uniform noise (Brendel et al., 2024), giving an approximate quantization function:

$$Q_L(x) \approx \tanh x + \frac{\mathcal{U}\{-1,1\}}{L-1} \tag{2}$$

which contains no explicit quantization. We also utilize use straight-through gradient estimation. We find that randomly mixing these two approaches along with unmodified latents produces better performance compared to utilizing only one or the other. This random mixing is achieved by starting with the unmodified latents, then replacing elements according to a random mask derived from a Bernoulli distribution with a parameter of $0.5$. This procedure is performed twice, once for elements with the straight-through approximation and once with the noise-based approximation. During training we also randomly select uniformly between a pre-selected set of quantization level numbers $L$. This is similar to the quantizer-dropout process used in training RVQ-based bottlenecks, and allows us to trade-off quality and codebook size at inference time.

### 3.2.1 Post-training bottleneck modification

The formulation of FSQ used here has many post-training possibilities for adjusting the reconstruction quality against the number and range of the discrete tokens. Firstly, the regularization provided by training the FSQ bottleneck with uniform noise allows the number of levels for each dimension of the FSQ to be modified after training. As long as the number of levels is greater than or equal to the smallest seen during training, the error produced by the quantization is within the bounds previously seen and therefore is still valid.

By default FSQ produces one token per time-step. In general this is advantageous for our purposes. However, if the use-case requires it, we can decompose this single token post-hoc into multiple tokens using either a parallel partitioning of the dimensions, or (for particular choices of quantization-level number) into a hierarchical residual set of tokens ala RVQ. Parallel partitioning introduces a bi-directional causal relationship between tokens which is unexplored in the generative modeling context, and therefore for this work we concentrate on the hierarchical residual decomposition.

Residual FSQ can be applied post-hoc to a bottleneck trained with a single quantizer but requires some restrictions. Namely, is required to only use numbers of levels conforming to $L = 2^n + 1, n \in \mathbb{Z}^+$. This sequence of levels can be derived by starting from levels at $\{-1, 0, 1\}$ ($L = 3$), and continually subdividing the intervals between levels exactly at the half way point. These level configurations are shown up to $n = 3$ in Tab. 1. We denote the set containing the positions corresponding to a particular number of levels $L$, as $\ell_L$. We can clearly see by examination that each larger set is a superset of the previous sets i.e $\ell_{2^n+1} \supset \ell_{2^{n-1}+1}$, and also that we can can construct any particular set of levels using the Minkowski sum of smaller $\ell_3$ sets, progressively halved e.g $\ell_3 + \frac{\ell_3}{2} \supset \ell_5, \ell_3 + \frac{\ell_3}{2} + \frac{\ell_3}{4} \supset \ell_9$ (albeit with extraneous new values outside the original range). A similar analysis holds for other level numbers conforming to the restriction given above, with the scalings consequently changed. We can utilize this property to do post-hoc residual quantization, using the standard formulation of a residual quantizer for a given latent $\mathbf{z}$:

$$\hat{\mathbf{z}} = \sum_{k=0}^{K} q_k,$$

$$\mathbf{q}_0 = \kappa_0(\mathbf{z}),$$

$$\mathbf{q}_k = \kappa_k(\mathbf{z} - \sum_{i=0}^{k-1} \mathbf{q}_i) \tag{3}$$

where $q_k$ denote the quantizer outputs, and $\kappa_k$ denote the quantizer functions themselves, which we define in terms of our scalar quantizer function using levels $L = 2^n + 1, n \in \mathbb{Z}^+, Q_{2n+1}$ as:

$$\kappa_k(\mathbf{z}) = \frac{Q_{2n+1}((2n)^k \mathbf{z})}{(2n)^k} \tag{4}$$

Using this formulation, we have the guarantee that the quantized latent $\hat{\mathbf{z}}$ belongs to the set of quantized levels seen during training, despite not having been trained using a residual formulation. A downside of this approach is that some rare combinations of tokens result in latents outside the bounds of those seen originally. This can be guarded against by clipping the output of the quantizer in the interval $[-1, 1]$.

As quantization noise training (Brendel et al., 2024) is used, it is also possible to remove the quantization entirely and use the latent as a continuous embedding, by retaining only the $\tanh$ section of the quantization function in which case the autoencoder operates as if it has a $\tanh$ bottleneck of the same latent dimension as the FSQ bottleneck.

| Quantized Positions | |
| --- | --- |
| $\ell_3$ | $\{-1, 0, 1\}$ |
| $\ell_5$ | $\{-1, -0.5, 0, 0.5, 1\}$ |
| $\ell_9$ | $\{-1, -0.75, -0.5, -0.25, 0, 0.25, 0.5, 0.75, 1\}$ |

Table 1: FSQ quantization points for level numbers conforming to $L = 2^n + 1$, $n \in \mathbb{Z}^+$, up to $n = 3$.

### 3.2.2 CALCULATING FSQ BITS-PER-SECOND

Using the post-hoc modification strategies described in Sec. 3.2.1, it is possible to achieve varying bits-per-second rates even for the same level of resolution.

We calculate bits-per-second (bps) for a decomposition with $n$ residual levels as:

$$\text{bps} = f_r \sum_{i=0}^{n} \lceil \log_2(k_i) \rceil \tag{5}$$

where $f_r$ is the number of frames per second of the codec (i.e. its latent rate) and the $k_i$ are the codebook sizes for each stage of the residual decomposition. We obtain these codebook sizes as:

$$k = L^d \tag{6}$$

where $L$ is the number of FSQ levels for the residual stage and $d$ is the FSQ dim.

For example, if we have an FSQ bottleneck with $L = 17$, $d = 6$, and a frame rate of 25Hz during training, this results in an effective bps of $25 \times \lceil \log_2(17^6) \rceil = 625$. If we partition this codebook into a residual formulation of 2 stages with 5 levels, we have an effective bps of $25 \times 2 \times \lceil \log_2(5^6) \rceil = 700$ but with a much more manageable codebook size for generative modelling. The same calculation of bitrate can be used for RVQ, using the chosen codebook size for each residual level.

### 3.3 DISCRIMINATOR

We employ a discriminator inspired by that used in Encodec (Défossez et al., 2022), consisting of multiple complex STFTs at different resolutions, followed by a combination of 1d and 2d convolutions. We make three major changes compared to previous versions of this discriminator: we scale parameter count by increasing the number of channels, we address systemic biases in the discriminator by adopting unevenly spaced STFT resolutions, and we address a late-training bias towards the noise-floor of the signal by scaling the magnitude of the complex STFTs before they are processed by the convolutional networks. The last two of these changes are motivated by analyzing the sensitivity of the discriminator to different regions of the input, and are justified in Appendix B.5.

### 3.4 TRAINING OBJECTIVES

Training the model is conducted in two stages with slightly different loss configurations - which we refer to as *pretraining* and *finetuning*. In each stage, the loss is a composite between several reconstruction losses and an adversarial loss derived from the discriminator network, which is trained in parallel. The main difference between pretraining and finetuning stages is in which reconstruction losses are used.

Similar to Défossez et al. (2024), we simplify the adversarial loss by removing the direct adversarial classifier loss term and using only a normalized feature-matching L1 loss on the $M$ per-layer features

of the multi-discriminator network containing $N$ individual discriminators, given by:

$$L_{\text{disc}}(\mathbf{x}, \hat{\mathbf{x}}) = \frac{1}{MN} \sum_{m=1}^{M} \sum_{n=1}^{N} \frac{\|D_n^m(\mathbf{x}) - D_n^m(\hat{\mathbf{x}})\|_1}{\text{mean}(\|D_n^m(\mathbf{x})\|_1)}, \tag{7}$$

where $D_n^m$ is the output of the $m$-th layer of the $n$-th individual discriminator, $\mathbf{x}$ is the target signal and $\hat{\mathbf{x}}$ is the reconstructed signal. This can be interpreted as a reconstruction loss using more semantically focused projections of the signal, which additionally adapt throughout the training as the discriminator improves. The discriminator is trained as usual as a binary classifier for real and fake examples utilizing a hinge loss.

During the pretraining stage, we include a traditional L1 reconstruction loss and L1 STFT loss to boost convergence. This loss is weighted by a coefficient that decays exponentially per-step. This ensures that the reconstruction loss does not influence the training after an initial period defined by the exponential decay factor. The overall loss during pretraining is given by:

$$L_{\text{pre}}(\mathbf{x}, \hat{\mathbf{x}}) = L_{\text{disc}}(\mathbf{x}, \hat{\mathbf{x}}) + \gamma^k L_1(\mathbf{x}, \hat{\mathbf{x}}) + \gamma^k L_1(|\mathbf{X}|, |\hat{\mathbf{X}}|) \tag{8}$$

where $\gamma$ is an exponential decay coefficient, $k$ is the training step and $\mathbf{X}, \hat{\mathbf{X}}$ are the bins of the STFT of the target and reconstructed signals respectively.

During the finetuning stage, we add a perceptual loss based on a pre-trained `WavLM-Large`(Chen et al., 2022) model. This perceptual loss is calculated similarly to discriminator feature-matching loss given in Eq. 7, by calculating L1 loss on the layer features of the target and reconstructed examples and normalizing by the mean magnitude of the target feature across the batch:

$$L_{\text{perc}}(\mathbf{x}, \hat{\mathbf{x}}) = \frac{1}{M} \sum_{m=1}^{M} \frac{\|C^m(\mathbf{x}) - C^m(\hat{\mathbf{x}})\|_1}{\text{mean}(\|C^m(\mathbf{x})\|_1)}), \tag{9}$$

where $C_m$ is the $m$-th layer of the model. We utilize all individual layer features supplied by the model. The overall loss during finetuning is given by:

$$L_{\text{fine}}(\mathbf{x}, \hat{\mathbf{x}}) = L_{\text{disc}}(\mathbf{x}, \hat{\mathbf{x}}) + L_{\text{perc}}(\mathbf{x}, \hat{\mathbf{x}}) \tag{10}$$

We found this finetuning stage to be essential in producing intelligible speech, as well as improving objective metrics, as shown in the ablation studies presented in Appendix A.1.

## 4 EXPERIMENTS

### 4.1 DATA

For training speech codec models, we use the Librilight dataset (60k hours) and the English portion of the Multilingual LibriSpeech (MLS) dataset (45k hours). Both datasets contain 16 kHz original speech data, amounting to a total of approximately 105k hours of training data. For evaluation, we utilize the `test-clean` subset of LibriSpeech for speech data, selecting audio clips with durations ranging from 5 to 10 seconds to create a test set of 900 clean speech samples at 16 kHz.

### 4.2 MODEL AND TRAINING DETAILS

The codec model is configured with a patch size of 320 samples at the input. There are two encoder blocks. One directly follows the patching, and contains 8 transformer blocks. This is followed by a further encoder block performing 2x downsampling, which contains 20 transformer blocks. The embedding dimension of the transformer blocks is 1024, whilst the reverse bottleneck of the feedforward layer is 4x larger. The head dimension of the self-attention block is 128. Layer norms are configured with $\epsilon = 1 \times 10^{-2}$, and the sliding attention window is of size 128. The decoder is configured to be symmetrical with the encoder. The resulting model has approximately 950M parameters. The bottleneck is 6 dimensional and trained with 17, 9 and 5 levels for every dimension, randomly chosen. The ensemble discriminator is configured as described in Appendix B.5, with each discriminator having a channel count of 256. We use FlashAttention (Dao et al., 2022) to ensure computational efficiency. The model is trained with `FP16` mixed precision.

The AdamW optimizer is used for both the autoencoder and discriminator, both with a learning rate of 0.0008. The autoencoder additionally uses weight decay with a coefficient of 0.01. Data is randomly chunked into segments of 5.12 seconds for training. 16 H100 GPUs are utilized, with an effective batch size of 128. Pretraining is conducted for 500k steps, with a decay coefficient of $\gamma = 0.9999$ applied to the reconstruction losses. The STFT loss utilizes 2048 bins, a hop size of 512 and a Hanning window. The finetuning stage is conducted for a further 150k steps using the `WavLM-Large` perceptual reconstruction loss in addition to the adversarial feature-matching loss. In both stages, all loss terms are weighted equally.

### 4.3 OBJECTIVE AND SUBJECTIVE METRICS

A set of objective metrics are used to assess perceptual quality, compression levels, reconstruction fidelity and semantic performance. These metrics are described in Appendix D.

We further conduct a subjective test with 24 participants that rate a total of 25 reconstructions from the same dataset used for objective metrics. We follow the precedent of previous works (Zhang et al., 2023b; Défossez et al., 2022) and employ the MUSHRA (Schoeffler et al., 2018) format without hidden anchor. Listeners compare multiple versions of an example at once, including a labeled reference and a hidden reference and are asked the question "*Please evaluate the quality proximity between an audio sample and its reference. Please listen carefully to the reference audio and then rate the quality of each test audio clip compared to the reference. Use the scale where 0 indicates no resemblance to the reference, and 100 means perfectly the same as the reference.*". Participants were gathered online by openly by sharing a link to the test in a number of public forums. To limit the length of the subjective test, we only select a subset of the baselines for inclusion. These are chosen based on overall performance on objective metrics vs bits-per-second. The demographic breakdown of the participants is shown in Appendix E.

### 4.4 BASELINES

We compare our results against the 16 kHz models DAC, SpeechTokenizer, and SemantiCodec, as well as the 24 kHz models Encodec and Mimi. For DAC, which produces speech tokens at 50 Hz, we use the first two or four levels of RVQ to achieve bitrates of 1 kbps and 2 kbps, respectively. SpeechTokenizer operates with the same token rate as DAC, and we retain the first two or three levels of EVQ to obtain bitrates of 1 kbps and 1.5 kbps. For SemantiCodec, we select the variant with a codebook size of 16, 384 and evaluate it at token rates of 25 and 50 per second, corresponding to bitrates of 340 bps and 680 bps. Encodec is evaluated at 1.5 kbps and 3 kbps. For Mimi, we use all 8 RVQ levels for a bitrate of 1.1 kbps, and the first 4 levels to achieve 550 bps. For the 24 kHz models Encodec and Mimi, we first upsample the test audio to 24 kHz for reconstruction and then downsample it back to 16 kHz for evaluation.

The baseline models differ in design goals and applications: DAC, Encodec, and SemantiCodec support diverse audio domains (multilingual speech, music, general audio); Mimi focuses on streaming efficiency; and SpeechTokenizer is English speech-specific. Parameter counts also vary widely. While our work focuses on speech coding with training and evaluation on English datasets, the aim is to demonstrate the feasibility of a transformer-based speech codec and its scalability to larger parameter counts. The comparison between these models is framed within this context, as they represent recently published audio codecs with strong performance in speech coding. Differences in streamability, training data, and model size are detailed in Tab. 12.

### 4.5 MAIN RESULTS

We evaluate two variations of our model, with different post-hoc configurations of the FSQ bottleneck. One variant utilizes a single token per step, utilizing 6 levels for each of the 6 dimensions. This leads to an effective codebook size of $6^6 = 46656$. The other variant uses the residual decomposition described in Sec. 3.2.1 to use two residual tokens per step, each with an effective codebook size of $5^6 = 15625$. The procedure for calculating the quoted bits-per-second is described in Sec. 3.2.2. We additionally show objective results of the model with the quantizer removed from the bottleneck, giving an indication of the performance of the model if used with a diffusion model.

| Model | BPS | TPF | TPS | SISDR ↑ | Mel ↓ | STFT ↓ | PESQ ↑ | STOI ↑ | MOSNet ↑ |
|---|---|---|---|---|---|---|---|---|---|
| DAC | 1000 | 2 | 100 | −6.51 | 1.49 | 1.76 | 1.64 | 0.75 | 2.77 |
|  | 2000 | 4 | 200 | −0.37 | 1.07 | 1.41 | 2.29 | 0.85 | 2.95 |
| Encodec | 1500 | 2 | 150 | −0.22 | 1.14 | 1.49 | 2.36 | 0.85 | 2.87 |
|  | 3000 | 4 | 300 | 2.77 | 0.95 | 1.33 | 2.84 | 0.90 | 2.98 |
| SpeechTokenizer | 1000 | 2 | 100 | −3.30 | 1.06 | 1.37 | 2.41 | 0.85 | 2.94 |
|  | 1500 | 3 | 150 | −1.33 | 0.91 | 1.25 | 2.70 | 0.88 | 3.10 |
| SemantiCodec | 337.5 | 2 | 25 | – | 1.20 | 1.53 | 2.21 | 0.79 | 3.24 |
|  | 675 |  | 50 | – | 0.98 | 1.32 | 2.65 | 0.86 | 3.29 |
| Mimi | 550 | 4 | 50 | −4.45 | 1.19 | 1.55 | 2.48 | 0.85 | 3.11 |
|  | 1100 | 8 | 100 | 2.20 | 0.94 | 1.31 | 3.01 | 0.90 | 3.24 |
| TAAE | 400 | 1 | 25 | 3.18 | 0.97 | 1.35 | 2.96 | 0.90 | 3.36 |
|  | 700 | 2 | 50 | 4.73 | 0.86 | 1.26 | 3.09 | 0.92 | 3.36 |
| + no quant. | – | – | – | 5.08 | 0.85 | 1.25 | 3.12 | 0.92 | 3.36 |

Table 2: Evaluation results for objective metrics on speech codec models. We do not report SI-SDR results for SemantiCodec, as it is a generative model that lacks precise temporal alignment.

Results of the evaluation with the proposed objective metrics are given in Tab. 2. The two variants of our proposed structure show increased performance against the baselines in all objective metrics, whilst also being amongst the lowest in terms of bits per second and tokens per second. The residual variant of our proposed model shows higher performance by these metrics compared to the single-token and lower bits-per-second variant, but not by a large margin. The variant with FSQ bottleneck removed, and hence continuous latents, shows modestly boosted performance.

The results of the MUSHRA subjective test, shown in Fig. 2 indicate that TAAE obtains state-of-the-art results outperforming, by a significant margin, recently published speech codecs. Importantly, the proposed model obtains results that are close to the ground truth. Comparing these evaluation results with the baseline model sizes shown in Tab. 12 indicates the potential of scaling transformer-based codec architectures to achieve new benchmarks in terms of speech quality and compression. Our audio examples are online for listening[1].

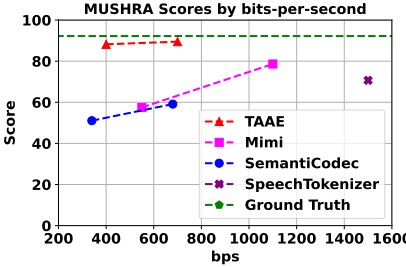

Figure 2: Results of MUSHRA test.

## 4.6 Additional Results

To evaluate the impact of model size, we conducted scaling experiments with TAAE architectures containing approximately 250M, 500M, and 1B parameters. The results confirm that the proposed structure scales effectively with parameter count, as detailed in Appendix A.2.

We also explored higher compression rates by modifying the encoder/decoder for $2\times$ additional up/downsampling (latent rate 12.5 Hz) and increasing the FSQ bottleneck dimension to $d = 8$. While this model achieves lower bitrates (e.g., 200 bps), it underperforms the main model and converges more slowly, as discussed in Appendix A.3.

In Appendix A.4, we describe and evaluate a causal version of the TAAE model. This variant shows minimal degradation compared to the non-causal version and outperforms the streaming codec Mimi in objective metrics, despite being trained with significantly fewer steps and data hours.

Additionally, we evaluated our proposed TAAE model across various settings beyond its original intended use case. In App. A.5, we assess the model performance on a range of languages, demonstrating its ability to generalize effectively to unseen languages, with results that are better or comparable to models trained on multiple languages. In App. A.6, we validate the model's generalization to utterances of varying lengths, including those longer or shorter than seen during training. We also compare our model with a HuBERT-based codec, analyzing key differences in design and performance, as discussed in Appendix A.7.

---

[1]https://stability-ai.github.io/stable-codec-demo/

Finally, we analyze the codebook utilization of TAAE, showing nearly optimal efficiency in its usage. A detailed comparison of codebook utilization and entropy-encoded bitrates across baselines is provided in Appendix A.8. A comparison of real-time factors between TAAE and the baselines is provided in Appendix A.9. This shows that despite the much larger parameter count, TAAE is competitive with baselines in terms of inference performance.

## 5 LIMITATIONS

The presented model has a number of limitations compared to baselines, primarily related to the training dataset rather than the architecture. We use only a modest amount of English-only speech data, 100k hours. The data is also at 16 kHz sampling rate, whereas 24 kHz or higher might be desirable in many applications. The dataset is predominantly made of audiobook recordings, so we might also expect the model to have difficulties with speech that is from a very different setting (e.g. overlapping speakers) or contains a significant amount of environmental sound. The limitations of the architecture are primarily related to parameter count and computational efficiency. The main presented model has a large parameter count which means that it may require greater computational resources than presented baselines, albeit mitigated by the availability of efficient transformer implementations. Future work should explore scaling up to a much larger and more diverse dataset at a higher sampling rate.

## 6 CONCLUSIONS

In this work we proposed a new scalable architecture for neural coding of speech waveforms, based on transformer-based encoder and decoder models and a flexible discrete bottleneck using Finite Scalar Quantization (FSQ). We described a number of techniques for altering or decomposing the tokens produced by this discrete bottleneck in order to fit the needs of various use-cases. We trained this architecture on a dataset of 16 kHz speech. We conducted objective and subjective evaluations of this model, showing state-of-the-art speech coding performance as well as generalization to unseen languages. The model can be adapted to streaming use-cases with little performance degradation, and is competitive with existing codec models in terms of inference speed, despite utilizing a much larger parameter count.

### ACKNOWLEDGMENTS

The authors would like to acknowledge the unpublished work of Dario Sucic on training-time residual FSQ formulations, shared in online discussions. We'd also like to acknowledge Shahbuland Matiana, with whom we had early discussions about transformer-based autoencoders, and Boris Kuznetsov with whom we had discussions about the use of noise in scalar quantizers.

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

# A    APPENDIX: ADDITIONAL RESULTS

We performed a number of additional experiments during the training process of the main presented model, the results of which are shown here.

## A.1    ABLATION ON FINETUNING USING PERCEPTUAL LOSS

Tab. 3 shows objective metrics for the main presented model, before and after the finetuning stage with using the `WavLM` perceptual feature-matching loss. As can be seen, this finetuning boosted sound quality metrics significantly, as well as significantly improving intelligibility – albeit at the cost of a tiny degradation in SI-SDR.

| Model | SI-SDR ↑ | Mel ↓ | STFT ↓ | PESQ ↑ | STOI ↑ |
|---|---|---|---|---|---|
| TAAE | 4.73 | 0.86 | 1.26 | 3.09 | 0.92 |
| w.o. perceptual loss | 4.80 | 1.18 | 1.59 | 2.82 | 0.88 |

Table 3: Evaluation results for the TAAE model at 700 BPS with and without perceptual loss.

## A.2    ABLATION STUDIES ON MODEL SCALING

To evaluate the effect of increasing model size on the performance of the TAAE architecture, we repeated the 500k step pretraining phase with models of approximately half and one quarter the parameter count of the main presented model. This is achieved by reducing the transformer embedding dimension to 768 and 512 respectively, whilst keeping all other hyper-parameters the same. Objective metrics for the trained models are shown in Tab. 4. We can see that scaling parameter count shows a clear improvement in objective metrics, although the smaller models still have respectable performance compared to baselines.

| Param. count | SI-SDR ↑ | Mel ↓ | STFT ↓ | PESQ ↑ | STOI ↑ |
|---|---|---|---|---|---|
| 240M | 3.52 | 1.24 | 1.67 | 2.74 | 0.87 |
| 540M | 4.31 | 1.21 | 1.66 | 2.80 | 0.88 |
| 950M | 4.80 | 1.18 | 1.59 | 2.82 | 0.88 |

Table 4: Evaluation results for TAAE model at 700 BPS with a variety of parameter counts.

## A.3    TRAINING MODELS WITH HIGHER COMPRESSION RATES

Tab. 5 shows the objective results of training the same architecture as our main presented model, with two major changes. The larger block in the encoder/decoder is split into two to provide an extra 2x upsampling/downsampling, giving an overall latent rate of 12.5 Hz. Additionally the dimension $d$ of the FSQ bottleneck is increased to 8. The parameter count is the same, apart from a minor difference in the layers mapping into and out of the bottleneck. This model performs worse than the presented model (as shown in Tab. 2) in most metrics, albeit operating at a much lower bit-rate. Observation during training showed that this model converged much slower than the presented model, so this gap might close with additional training.

| Latent (Hz) | FSQ (L × d) | BPS | TPF | TPS | SI-SDR ↑ | Mel ↓ | STFT ↓ | PESQ ↑ | STOI ↑ |
|---|---|---|---|---|---|---|---|---|---|
| 12.5 | 4 × 8 | 200 | 1 | 12.5 | −1.40 | 1.26 | 1.61 | 2.34 | 0.82 |
| | 5 × 8 | 325 | 2 | 25 | 0.58 | 1.13 | 1.49 | 2.49 | 0.84 |
| | 9 × 8 | 488 | 3 | 37.5 | 2.56 | 1.05 | 1.42 | 2.66 | 0.87 |
| | 17 × 8 | 650 | 4 | 50 | 3.37 | 1.02 | 1.40 | 2.73 | 0.88 |
| 25 | 6 × 6 | 400 | 1 | 25 | 3.18 | 0.97 | 1.35 | 2.96 | 0.90 |
| | 17 × 6 | 700 | 2 | 50 | 4.73 | 0.86 | 1.26 | 3.09 | 0.92 |

Table 5: Objective results for proposed speech codec models with different latent rate.

## A.4  CAUSAL MODEL VARIATION FOR STREAMING USE

Although the purpose of this work was not to produce an audio codec intended for streaming use, it is possible to make some small modifications to the structure to make it fully causal. Firstly we shift the sliding attention window so that it is purely causal, instead of symmetrical around the query. Secondly, we replace the non-causal convolution layers with causal implementations.

In order to test the impact of these design changes, we finetune a fully trained non-causal model with these causal modification for 200k steps using the same objective as the finetuning phase of the main model. Objective metrics for this model are shown in Tab. 6. We can see that the causal version of the model performs marginally worse in terms of objective metrics, but is still competitive with Mimi, which is the strongest baseline trained with 4 million steps and 7 million hours of speech data. Note that if designing a model ground-up for streaming use, it may be wise to choose a smaller parameter count in order to meet a specific real-time compute budget. The results in Appendix A.2 suggest that smaller models may be viable in this use-case.

In fully causal mode, the latency of the model is dictated by the latent rate. The model presented here has a latent rate of 25Hz, resulting in a latency of 40ms.

| Model | BPS | SI-SDR ↑ | Mel ↓ | STFT ↓ | PESQ ↑ | STOI ↑ | MOSNet ↑ |
|---|---|---|---|---|---|---|---|
| Mimi | 1100 | 2.20 | 0.94 | 1.31 | 3.01 | 0.90 | 3.24 |
| TAAE (causal) | 700 | 4.04 | 0.94 | 1.31 | 3.09 | 0.92 | 3.34 |
| TAAE (non-causal) | 700 | 4.73 | 0.86 | 1.26 | 3.09 | 0.92 | 3.36 |

Table 6: Evaluation results for the TAAE model at 700 BPS, before and after causal finetune.

## A.5  GENERALIZATION ON MULTILINGUAL SPEECH DATASETS

We assess the generalization capability of the TAAE model using the Multilingual LibriSpeech (MLS) dataset (Pratap et al., 2020). For this evaluation, we randomly select 500 audio clips for each non-English language: Italian, Polish, Dutch, French, German, Spanish and Portuguese. Each clip is between 5 and 10 seconds in duration and sampled at 16 kHz. TAAE is tested at 700 bps, with its performance compared against baseline audio codecs at low bite-rates. Objective metrics from Table 2 are used, except for MOSNet, which is excluded due to its training on English-only data. The results of this evaluation are detailed in Table 7.

Despite being trained exclusively on English speech data, TAAE generalizes effectively to multilingual datasets, consistently outperforming codecs trained with multilingual data, such as Encodec, DAC, and SemantiCodec, as well as SpeechTokenizer, another English-only model, across all objective metrics for all evaluated languages. Additionally, when compared to Mimi, which utilizes a massive dataset of 7 million hours of predominantly English speech (approximately 700 times larger than the training set of TAAE), TAAE achieves better performance on SI-SDR, Mel Distance, and STFT Distance, matches performance on STOI, and is only slightly underperformed on PESQ. These results highlight TAAE's ability to generalize to unseen languages despite its English-only training, suggesting its potential for even greater performance when trained on multilingual data, making it a promising solution for a wide range of multilingual applications.

## A.6  LENGTH GENERALIZATION

In Fig. 3 we show results from evaluating the presented model and baselines on utterances from the `test-clean` subset of LibriSpeech binned into categories at a variety of different lengths. Each length bin consists of utterances ±1s from the stated value. We see that all models handle inference at various lengths fairly gracefully. The TAAE clearly shows better performance around the 5s length it was trained on and degrades slightly with longer utterances. Similarly, Mimi has its best performance at the longer utterances length it was trained on, and degrades slightly with shorter utterances. Models using chunked inference exhibit the least performance variation with longer input segments, which aligns with expectations.

| Model | BPS | SI-SDR ↑ | Mel ↓ | STFT ↓ | PESQ ↑ | STOI ↑ |
|---|---|---|---|---|---|---|
| **Italian** | | | | | | |
| Encodec | 1500 | 0.63 | 1.20 | 1.55 | 2.40 | 0.85 |
| DAC | 2000 | −0.13 | 1.11 | 1.46 | 2.23 | 0.84 |
| SemantiCodec | 675 | — | 1.05 | 1.41 | 2.57 | 0.84 |
| SpeechTokenizer | 1000 | −2.61 | 1.07 | 1.42 | 2.40 | 0.84 |
| Mimi | 1100 | 2.69 | 1.02 | 1.42 | 3.00 | 0.90 |
| TAAE | 700 | 4.54 | 0.99 | 1.38 | 2.89 | 0.89 |
| **Polish** | | | | | | |
| Encodec | 1500 | 1.39 | 1.12 | 1.49 | 2.42 | 0.86 |
| DAC | 2000 | 1.30 | 1.02 | 1.40 | 2.38 | 0.87 |
| SemantiCodec | 675 | — | 1.08 | 1.42 | 2.36 | 0.85 |
| SpeechTokenizer | 1000 | −1.70 | 1.08 | 1.42 | 2.36 | 0.85 |
| Mimi | 1100 | 2.68 | 1.04 | 1.46 | 2.82 | 0.90 |
| TAAE | 700 | 4.45 | 0.95 | 1.36 | 2.66 | 0.89 |
| **Dutch** | | | | | | |
| Encodec | 1500 | 1.18 | 1.13 | 1.51 | 2.59 | 0.86 |
| DAC | 2000 | 1.30 | 0.98 | 1.36 | 2.55 | 0.87 |
| SemantiCodec | 675 | — | 1.09 | 1.42 | 2.34 | 0.83 |
| SpeechTokenizer | 1000 | −5.01 | 1.09 | 1.42 | 2.34 | 0.83 |
| Mimi | 1100 | 2.84 | 0.98 | 1.39 | 3.01 | 0.90 |
| TAAE | 700 | 4.03 | 0.90 | 1.29 | 2.93 | 0.88 |
| **French** | | | | | | |
| Encodec | 1500 | 3.12 | 1.16 | 1.50 | 2.51 | 0.85 |
| DAC | 2000 | 2.68 | 0.98 | 1.34 | 2.41 | 0.87 |
| SemantiCodec | 675 | — | 1.02 | 1.36 | 2.54 | 0.83 |
| SpeechTokenizer | 1000 | −0.50 | 1.04 | 1.36 | 2.38 | 0.84 |
| Mimi | 1100 | 4.61 | 0.98 | 1.38 | 2.98 | 0.89 |
| TAAE | 700 | 6.70 | 0.94 | 1.30 | 2.87 | 0.88 |
| **Portuguese** | | | | | | |
| Encodec | 1500 | −0.46 | 1.18 | 1.56 | 2.49 | 0.84 |
| DAC | 2000 | −1.05 | 1.07 | 1.44 | 2.35 | 0.84 |
| SemantiCodec | 675 | — | 1.04 | 1.42 | 2.59 | 0.83 |
| SpeechTokenizer | 1000 | −4.15 | 1.07 | 1.42 | 2.43 | 0.83 |
| Mimi | 1100 | 1.45 | 0.98 | 1.42 | 3.04 | 0.89 |
| TAAE | 700 | 3.14 | 0.93 | 1.33 | 2.93 | 0.87 |
| **German** | | | | | | |
| Encodec | 1500 | 0.04 | 1.17 | 1.53 | 2.40 | 0.84 |
| DAC | 2000 | −0.53 | 1.09 | 1.44 | 2.34 | 0.85 |
| SemantiCodec | 675 | — | 1.07 | 1.43 | 2.31 | 0.83 |
| SpeechTokenizer | 1000 | −3.86 | 1.10 | 1.43 | 2.31 | 0.83 |
| Mimi | 1100 | 1.84 | 1.01 | 1.42 | 2.95 | 0.89 |
| TAAE | 700 | 4.94 | 0.92 | 1.32 | 2.83 | 0.88 |
| **Spanish** | | | | | | |
| Encodec | 1500 | 2.32 | 1.21 | 1.54 | 2.42 | 0.86 |
| DAC | 2000 | 1.93 | 1.04 | 1.39 | 2.36 | 0.86 |
| SemantiCodec | 675 | — | 1.04 | 1.39 | 2.52 | 0.84 |
| SpeechTokenizer | 1000 | −0.84 | 1.07 | 1.42 | 2.43 | 0.85 |
| Mimi | 1100 | 3.82 | 1.07 | 1.44 | 2.93 | 0.90 |
| TAAE | 700 | 6.15 | 0.98 | 1.37 | 2.80 | 0.89 |

Table 7: Evaluation results of objective metrics on the Multilingual LibriSpeech (MLS) dataset.

## A.7 COMPARISON WITH HUBERT-BASED CODEC

We compare our approach with an alternative family of speech codecs that leverage discrete (semantic) tokens derived from self-supervised pre-trained speech models (e.g., HuBERT (Hsu et al., 2021)). These tokens are subsequently used by a generative model to resynthesize the waveform. In this study, we employ the pre-trained unit-based HiFi-GAN vocoder (Kong et al., 2020) model (unitHiFi-GAN), as used in SpeechGPT[2], to resynthesize waveforms discrete tokens from `HuBERT-base` model (95 M). The unitHiFi-GAN operates on HuBERT representations with a latent rate of 50 Hz for 16 kHz speech signals and utilizes a k-means clustered codebook with 1000 entries, resulting in an effective bitrate of 500 bps for 16 kHz speech. We apply unitHiFi-GAN to resynthesize the audio in the test set and report objective metrics to compare its performance with that of our proposed TAAE models. Results are shown in Table 8. We observe that unitHiFi-

---

[2]`https://github.com/0nutation/SpeechGPT/blob/main/speechgpt/utils/vocoder/`

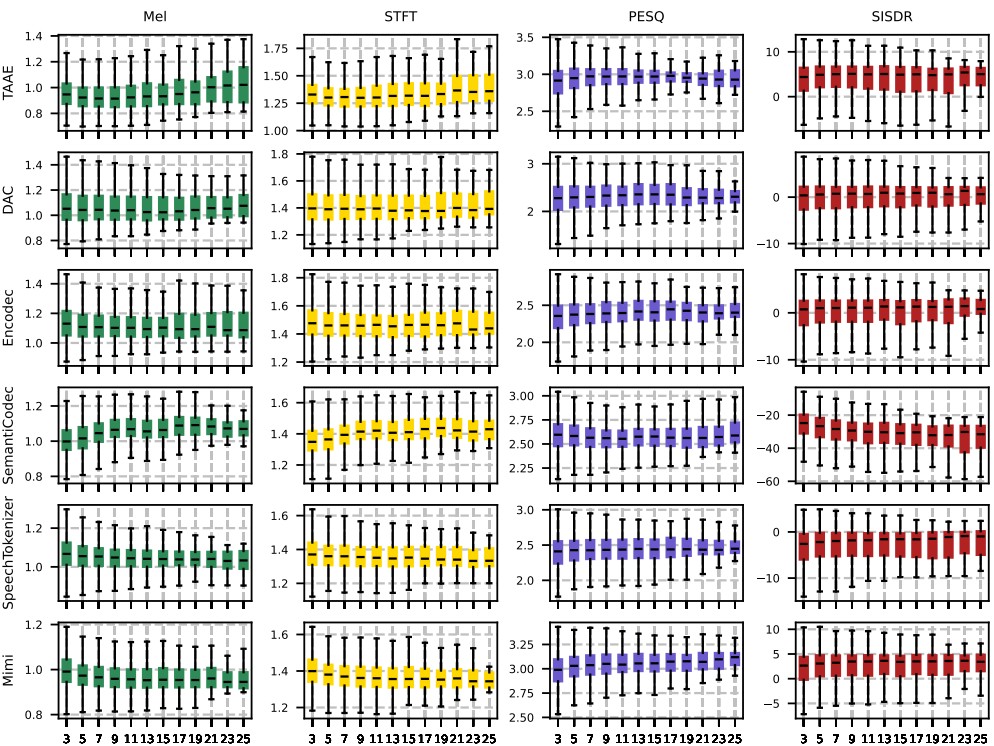

Figure 3: Objective metrics for the TAAE and baselines, evaluated on utterances from length 3s to 25s, showing generalization of models across lengths. In cases where a baseline has multiple bitrate versions evaluated in this work, the higher bitrate is evaluated here.

| Model | BPS | SI-SDR ↑ | Mel ↓ | STFT ↓ | PESQ ↑ | STOI ↑ | MOSNet ↑ |
|-------|-----|----------|-------|--------|--------|--------|----------|
| unitHiFi-GAN | 500 | −45.95 | 3.14 | 3.24 | 1.12 | 0.16 | 2.98 |
| TAAE | 400 | 3.18 | 0.97 | 1.35 | 2.96 | 0.90 | 3.36 |
| TAAE | 700 | 4.73 | 0.86 | 1.26 | 3.09 | 0.92 | 3.36 |

Table 8: Comparison with HuBERT-based codec model.

GAN performs poorly across all metrics when compared with TAAE at both 400 bps and 700 bps. Although the resynthesized audio achieves an acceptable perceptual quality with a MOSNet score of 2.98, it exhibits poor performance on critical metrics such as SI-SDR and Mel/STFT distance. This suggests that HuBERT's discrete tokens fail to preserve sufficient acoustic detail, resulting in reconstructed audio that does not have close alignment with the reference audio.

The HuBERT-based codec approach benefits from the low bitrate of speech tokens and the semantic (e.g., phonemic) representations learned through self-supervised objectives. However, it may introduce trade-offs, such as the loss of acoustic details and timbre in the resynthesized audio (Mousavi et al., 2024). These limitations can be mitigated by incorporating discrete tokens derived from additional speech models, such as those for pitch tracking and speaker classification (Polyak et al., 2021). In contrast, the scope of this work is to explore an end-to-end trained waveform codec model that achieves high-quality reconstruction while maintaining low-bitrate compression.

## A.8    ANALYSIS OF CODEBOOK UTILIZATION AND ENTROPY-CODED BITRATES

This section examines the codebook utilization and entropy-coded bitrates of our proposed TAAE models alongside baseline audio codecs. The evaluation is conducted using the LibriSpeech `train-clean-360` set, which consists of 104140 speech clips totaling 360 hours of audio.

To measure codebook utilization, we compute the Normalized Entropy (Thomas & Joy, 2006), defined as:

$$\text{Normalized Entropy} = -\frac{1}{\log_2(N)} \sum_{x=1}^{N} p(x) \log_2(p(x)),$$

where $N$ is the size of the codebook, and $p(x)$ represents the probability of each codebook index $x$. The normalization factor, $\frac{1}{\log_2(N)}$, ensures the entropy is scaled by the maximum possible entropy of the codebook, $\log_2(N)$, resulting in a value within the range $[0, 1]$. Higher values of normalized entropy indicate more efficient and uniform codebook utilization.

We use Huffman coding (Huffman, 1952) as an entropy coding method to compress the output token stream by assigning shorter codes to more frequently occurring tokens, thereby minimizing the average bitrate. To compute the Huffman-coded bitrate values, we first determine the probability distribution $p(x)$ of each codebook index based on its frequency of occurrence in the dataset. Using these probabilities, Huffman coding assigns variable-length binary codes to each index, minimizing the average bitrate. The Huffman-coded bitrate is then calculated as:

$$\text{Huffman-Coded Bitrate} = \sum_{x=1}^{N} p(x)l(x),$$

where $l(x)$ is the length of the Huffman code assigned to index $x$. This value represents the compressed bitrate achieved after entropy coding.

The results presented in Table 9 show the codebook utilization and Huffman coding efficiency of the evaluated codec models. Encodec exhibits relatively low normalized entropy (0.78–0.86), indicating suboptimal codebook utilization. In contrast, recent RVQ-based approaches, such as DAC and Mimi, achieve significant improvements through techniques like Exponential Moving Average (EMA), factorized codes, and L2-normalized codes (Yu et al., 2022), reaching normalized entropy values of up to 0.99 and 0.92, respectively, across multiple RVQ levels. FSQ and residual FSQ (RFSQ), used in TAAE, achieve near-perfect utilization, with normalized entropy reaching 0.97.

The performance of Huffman coding correlates with codebook utilization. Models with lower normalized entropy benefit more from entropy coding due to their skewed token distributions. For example, Encodec achieves a 16.67% reduction in bitrate, from 3000 bps to 2510 bps. In contrast, our proposed TAAE and recent RVQ-based models, such as SpeechTokenizer and Mimi, exhibit smaller gains from entropy coding (e.g., TAAE achieves a reduction from 700 bps to 670 bps). This reflects a design choice: these models prioritize efficient codebook utilization to enhance reconstruction quality, limiting the potential for additional bitrate reduction through entropy coding.

### A.9 COMPARISON OF REAL-TIME FACTOR PERFORMANCE BETWEEN TAAE AND BASELINES

To evaluate the real-time performance of different audio codec models, Real-Time Factor (RTF) values were computed for three audio durations: 5 seconds, 30 seconds, and 60 seconds. Each test set consisted of 1000 audio clips, ensuring a robust assessment. The experiments were conducted on an NVIDIA H100 GPU. The RTF values indicate the processing speed relative to real-time playback, with lower values denoting faster processing. The results are shown in Tab. 10. We can see that the TAAE is competitive with the baselines in terms of RTF, despite a much larger parameter count. This can largely be attributed to a combination of the availability of highly optimized transformer implementations and the efficiency gains of offloading the majority of downsampling/upsampling to the very computationally efficient patched transform.

## B   APPENDIX: ADDITIONAL ARCHITECTURE DISCUSSION

### B.1   LAYER NORMALIZATION IN TRANSFORMERS WITHOUT TRAINABLE EMBEDDINGS

In contrast to the standard use-cases of transformers which involve discrete inputs with trainable embeddings, our architecture must deal with diverse inputs derived directly from signals. This includes

| Model | Codebook | Codebook Size | Normalized Entropy | BPS | Huffman-Coded |
|---|---|---|---|---|---|
| DAC | RVQ-1 | | 0.87 | | |
| | RVQ-2 | 1024 | 0.97 | 2000 | 1900 |
| | RVQ-3 | | 0.97 | | |
| | RVQ-4 | | 0.99 | | |
| Encodec | RVQ-1 | | 0.78 | | |
| | RVQ-2 | 1024 | 0.84 | 3000 | 2510 |
| | RVQ-3 | | 0.85 | | |
| | RVQ-4 | | 0.86 | | |
| SpeechTokenizer | RVQ-1 | | 0.98 | | |
| | RVQ-2 | 1024 | 0.95 | 1500 | 1450 |
| | RVQ-3 | | 0.95 | | |
| SemantiCodec | Semantic | 16384 | 0.86 | 675 | 625 |
| | Acoustic | 8192 | 0.99 | | |
| Mimi | RVQ-1 | | 0.91 | | |
| | RVQ-2 | | 0.90 | | |
| | RVQ-3 | | 0.90 | | |
| | RVQ-4 | 2048 | 0.91 | 1100 | 1005 |
| | RVQ-5 | | 0.92 | | |
| | RVQ-6 | | 0.92 | | |
| | RVQ-7 | | 0.92 | | |
| | RVQ-8 | | 0.92 | | |
| TAAE (400 BPS) | FSQ-1 | 46656 | 0.97 | 400 | 378 |
| TAAE (700 BPS) | RFSQ-1 | 15625 | 0.97 | 700 | 670 |
| | RFSQ-2 | | 0.95 | | |

Table 9: Comparison of codebook utilization and Huffman coding performance across different codec models and configurations.

| Model | Parameters (M) | RTF (5s) | RTF (30s) | RTF (60s) |
|---|---|---|---|---|
| DAC | 76 | 0.0048 | 0.001 | 0.0008 |
| Encodec | 14 | 0.0049 | 0.0025 | 0.0024 |
| SpeechTokenizer | 104 | 0.0038 | 0.0024 | 0.0024 |
| SemantiCodec | 507 | 1.2327 | 0.8111 | 0.7085 |
| Mimi | 80 | 0.0047 | 0.0007 | 0.0003 |
| TAAE | 950 | 0.0143 | 0.0024 | 0.0013 |

Table 10: Real-Time Factors (RTFs) for audio codec models on test audio clips of 5s, 30s, and 60s duration using an H100 GPU.

very low amplitude embeddings corresponding to silent sections of audio. This is a particular challenge when combined with sliding-window attention, as the entire context of a particular attention block could consist of low-level channel or rounding noise which has been amplified to high levels by the layer-norm block. In early experiments, we found that this problem prevented convergence of the architecture by causing instability and poor gradients if a batch contained too much silence. This can be mitigated by stripping silence from the dataset, but such an approach does not produce a model robust to real-world use-cases that may indeed contain silence. Instead we raise the $\epsilon$ constant used in the calculation of normalization factors in the layer norm blocks, effectively capping the maximum factor by which the layer norm can amplify an embedding. This allows embeddings corresponding to silence to remain at a very low level, and allows convergence of the architecture. The exact appropriate value for $\epsilon$ is data-dependent and related to the average noise floor of the

data. In this work, we use $\epsilon = 1 \times 10^{-2}$, whereas the default $\epsilon$ value in PyTorch LayerNorm[3] is $1 \times 10^{-5}$. This choice caps the maximum amplification to 40 dB, instead of 100 dB for the default setting. To contextualize these values, the total dynamic range of 16-bit fixed-point audio is 96 dB, which means at the default LayerNorm settings even rounding noise at the input could be amplified to full-scale embeddings.

## B.2 MODEL CONTEXT LENGTHS, CAUSALITY AND LATENCY

Any layer that processes information across sequence steps has a particular context length or *receptive field*. In the case of a convolutional layer, this receptive field is finite and defined by the kernel size of the convolution as well as any striding or dilation. In the case of RNNs and self-attention the receptive field is limited only by the length of the data. Adding a sliding-window mask to self-attention limits the context to be finite like a convolutional layer. The receptive field of a complete network is the sum of the receptive fields of the individual layers, as each layer is potentially able to pass information from one extreme of it's per-layer receptive field to the other.

Correctly choosing the receptive field is a key design parameter in networks that must be generalized over different sequence lengths. If it's chosen to be too short, the network may not learn important long-term dependencies in the data. If it's chosen to be too long, the network can fail to generalize both to shorter sequences (if the whole context is needed for proper operation) and to longer sequences (if during training the network never sees a sequence exceeding its receptive field). This choice is especially crucial on a per-layer basis.

Related to the topic of receptive field is the concept of *streamability*. A streamable codec model could be used for live encoding and decoding of an audio stream, in a real-time operation context. To be viable for this operation mode, a model needs to have a reasonable *latency* i.e. delay between audio input and output. The simplest way of achieving low latency is to design a model which is strictly causal. The latency of the model is then dictated by the lowest temporal sampling frequency within the model, which is usually the latent rate. Alternatively, non-causal models may be made streamable by employing *chunked inference*. This means that the input signal is split up into (usually fixed size) chunks, with a certain amount of overlap to reduce boundary issues. In this case the latency is equal to two chunks. This is usually much too high for real-time applications.

### B.2.1 COMPARISON BETWEEN BASELINES

Our proposed TAAE model uses a sliding attention window of 128 steps in the main presented non-causal version, giving it a maximum per-layer receptive field 5.12s and an overall receptive field of approximately 200s, centered on the current time-step. The causal variation described in App.A.4 uses a causal sliding attention window of 64 steps, giving it a maximum per-layer receptive field of 2.56s and an overall receptive field of approximately 100s. When configured in a causal fashion, the streaming latency is 40ms.

Mimi (Défossez et al., 2024) uses a causal attention window of 250 steps, which given the attention happens only on 12.5Hz latents would give it a maximum per-layer receptive field of 20s and a total receptive field approximately 320s. It is capable of streaming with a latency of 80ms. Length generalization is not addressed in detail but is said to be acceptable up to 5mins, which is consistent with the total receptive field.

Encodec (Défossez et al., 2022) employs an RNN as part of its design, giving effectively unlimited per-layer and total receptive fields. The potential negative impact on length generalization during inference is mitigated by encoding and decoding audio in fixed-size chunks of 1s using overlap, when in non-streaming mode. In streaming mode, this potential negative impact is not addressed.

DAC (Kumar et al., 2023) is purely convolutional, so has a much smaller receptive field than the other models considered here. Its maximum per-layer receptive field is 0.16s, whilst its total receptive field is 0.76s. It is not causal, but its limited receptive field means that it could be used for streaming purposes with a latency of 0.38s.

SemantiCodec (Liu et al., 2024a) uses both RNNs and attention without a sliding window, giving it effectively unlimited per-layer and total receptive fields. It is trained on 10s segments, and inference

---
[3]https://pytorch.org/docs/stable/generated/torch.nn.LayerNorm.html

is also chunked to this length with overlap. It is non-causal, but streaming is possible with a large latency due to the chunked inference.

SpeechTokenizer (Zhang et al., 2023b) uses RNNs, giving it effectively unlimited per-layer and total receptive fields. It is trained on 3s segments. The authors do not discuss if chunked inference is used to aid length generalization or to allow streaming.

All models are expected to effectively scale as $\mathcal{O}(n)$ with sequence length, due to the use of limited attention context windows or chunked inference. We summarize the above information in Tab. 11.

| Model | Max Per-Layer RF | Total RF | Causal | Latency | Chunked Inference |
|-------|------------------|----------|--------|---------|-------------------|
| Encodec | Unlimited | Unlimited | Yes | 13ms | Optional |
| DAC | 0.16s | 0.76s | No | 0.38s (non-chunked) | Optional |
| SemantiCodec | Unlimited | Unlimited | No | 20s | Yes |
| SpeechTokenizer | Unlimited | Unlimited | No | N/A | Not discussed |
| Mimi | 20s | $\sim 320$s | Yes | 80ms | No |
| TAAE (Non-Causal) | 5.12s | $\sim 200$s | No | N/A | No |
| TAAE (Causal) | 2.56s | $\sim 100$s | Yes | 40ms | No |

Table 11: Comparison of receptive field and streamability across different models. RF stands for the receptive field. Unlimited RF values do not take into account chunked inference.

## B.3 CONVOLUTION VS ATTENTION

The most fundamental change compared to previous codec models in our architecture is the switch from a predominantly convolutional architecture to one that closely resembles a standard transformer. As discussed above, this is predominantly motivated by the success of scaling such architectures in other fields (Hoffmann et al., 2022; Dosovitskiy et al., 2021). This switch represents a move from an architecture with strong inductive bias, high parameter efficiency but poor scaling, to an architecture that is more general, less parameter efficient, but has more potential to scale. Beyond these high-level differences, it is interesting to also explore the difference between convolution and attention, and how this motivates the change.

Convolution and attention are actually more similar than it would first appear. Attention maps, being weighted sums over an input, are essentially a 1d convolution kernel. Compared to the convolution kernels learned in a typical 1d convolution layer, however, they have four major differences. Firstly, attention maps generally cover a much larger receptive field than learned convolution kernels. Secondly, the nature of the $\mathrm{softmax}$ function used to calculate the attention map restricts the individual weights to be purely positive and sum to one (imparting an overall lowpass characteristic to the resulting kernel). Thirdly, convolution kernels are usually learned and applied in a dense format or per-channel, whereas attention maps are applied uniformly over a projection of the channels (an attention head). Lastly, the attention maps can vary dynamically with input and are therefore not restricted to be shift-invariant (although they may learn to be). We would argue that even with the lowpass constraint and the restriction of attention heads, attention can effectively be thought of as a superset of the capabilities of convolution.

The relaxation of the shift-invariant constraint in particular could be useful for a model intended to perform information compression. A particular audio input sequence does not necessarily have an even distribution of information along the sequence, for example silent segments will contain much less information than active speaking. Intuitively using purely a shift-invariant operation would make it harder to address this difference in information density, given that it acts uniformly across the sequence. In contrast, attention is able to freely move information around and utilize the whole length of the sequence as it see's fit – potentially allowing it to dedicate more capacity to the information-dense portions of the input signal and less to the information-sparse portions.

## B.4 FILTERBANK CHOICE FOR CODEC DESIGN

The above described architecture relies on using a fixed non-learnable transform on the raw audio waveforms to perform a large proportion of the temporal downsampling and channel expansion.

This type of transform is known in the signal processing literature as a filter-bank or time-frequency representation (Smith, 2011). The most common example of such a transform is the Short Time Fourier Transform (STFT).

When choosing an appropriate transform, there is a number of important qualities to consider. Firstly there is *perfect-reconstruction* – this means that an exact inverse of the transform exists which can recover the original waveform. Secondly there is *critical-sampling* – which means the number of channels of the transform matches the temporal down-sampling and hence does not expand the amount of data. Thirdly, there is the resistance of these transforms to manipulation and error. This property is mostly effected by how heavily the transform relies on *time-domain aliasing cancellation* (TDAC) to counteract the effect of the aliasing produced by downsampling each channel. TDAC requires very specific relationships between channels to be maintained in order to work, which may be perturbed by reconstruction error resulting in errors in the final waveform.

We now describe some of the transforms that were considered during the development of this model.

**STFT** – The STFT only fulfills both the perfect-reconstruction and critical-sampling criteria in a single case - when a rectangular window with no overlap is used. In this configuration the transform is highly sensitive to errors, which generally manifest themselves as periodic transients at the boundaries between STFT frames. If the critical-sampling requirement is relaxed and windowing with the correct overlap is used, an STFT may have both perfect-reconstruction and excellent error resistance, however achieving this generally expands the length of the input sequence very significantly.

**MDCT** – The Modified Discrete Cosine Transform (MDCT) is used in many traditional audio coding algorithms. It is both critically sampled and possesses perfect reconstruction. In practice we found that, like the rectangular-windowed STFT, the artefacts resulting from error before reconstruction were periodic and perceptually undesirable.

**PQMF** – The Pseudo Quadrature Mirror Filter (PQMF) approach is also used it traditional audio coding algorithms. It is critically sampled, but does not conform exactly to the perfect-reconstruction criteria. Its error resistance is fairly strong, as TDAC only happens between adjacent channels in the representation.

**Patched transform / Polyphase** – In the signal-processing literature, the patched transform is generally known as a polyphase filterbank (Smith, 2011). It is an edge-case in the world of transforms, in that each channel covers the same frequency band, but with a different phase offset. It conforms to both the perfect-reconstruction and critically-sampled criteria, and is often used for efficient filter-implementation in the signal processing literature for this reason. The polyphase transform looks at first glance to have poor resistance to error, given that TDAC happens between all channel simultaneously. However, in practice this means that the effect of the error after reconstruction is much more evenly distributed across both time and frequency, lacking the periodic elements seen with other transforms. We found that this made reconstruction error perceptually much improved, and consequently chose the patched / polyphase transform as the appropriate transform for this model.

### B.5 SYSTEMATIC BIAS IN LOSS FUNCTIONS

During the initial experiments leading to the work described here, we noticed a tendency for the presented architecture to produce consistent periodic artifacts (seen as lines in a spectrogram), especially in the upper frequencies of a sound. These artifacts often disappeared with additional training, but not consistently. One theory for their origin was that the increased capacity of the proposed architecture compared to previous codec models encouraged the model to overfit on biases in the training objective.

In order to examine this theory, we can define a per-sample sensitivity metric for any particular loss $L(\mathbf{x})$, with respect to the input signal samples $x_n$ of the input signal $\mathbf{x}$:

$$s_n = \left| \frac{\partial L(\mathbf{x})}{\partial x_n} \right| \tag{11}$$

This metric can be extracted easily from a given neural network structure and an example signal, using automatic differentiation.

In addition, if we want a more detailed view of the sensitivity of the loss to any particular frequency element of the input signal, we can first perform a time-frequency transformation like an STFT to

the signal, apply the inverse transformation before processing with the network, and calculate the derivative with respect to the bins $X_{n,f}$ of the transformed signal:

$$s_{n,f} = \left| \frac{\partial L(\mathbf{x})}{\partial x_{n,f}} \right| \tag{12}$$

To measure bias using this metric, we average the $S_{n,f}$ values over many different example inputs.

We first conducted this analysis on common reconstruction loss metrics including L1 loss, L2 loss and STFT-based losses. All showed no systematic bias. In the STFT case, this is predicated on correct windowing and overlap of the STFT to fulfill the requirements for perfect reconstruction.

However, performing this analysis on the feature-matching, adversarial and discriminator losses used for adversarial training revealed clear systematic bias. A freshly initialized Encodec discriminator using power-of-two FFT sizes and overlaps, as is standard, produced clear horizontal and vertical lines in the sensitivity spectrum. Each of these set of lines appeared to be connected to a single STFT discriminator. This indicated that the the gradients used to train both the discriminator and the codec were biased towards particular time-steps and towards particular frequencies. This analysis was repeated on a fully trained discriminator network. The training process somewhat mitigated this bias, but clear horizontal and vertical lines were still present. We postulated that this was the source of the periodic artifacts in the reconstructions from the model. Similar behaviour was seen in the discriminator design from DAC (Kumar et al., 2023) and BigVGAN (Lee et al., 2022), with DAC seeming to be suffer particularly from periodic artifacts due to the multi-period part of its discriminator. A deeper examination of the reason why this bias effects a transformer-based architecture more than previous convolutional architectures is left to future work.

As the regular patterns in the sensitivity appeared to be related to the FFT sizes used, an attempt was made to mitigate using techniques inspired by older work in artificial reverberation. When designing an artificial reverberator, one of the main challenges is to make sure that the regular spectral peaks produced by comb or allpass filters do not coincide and cause metallic ringing. One strategy is to tune these filters to be maximally inharmonically related. We achieve this for FFT sizes, by choosing a base FFT hop size and then generating a number of new hop sizes by multiplying with a constant interval. The FFT sizes are then chosen to be double this hop size, to maintain perfect reconstruction for the used Hanning window. Using an optimization procedure, the constant interval that produced the most inharmonic relationship was found to be close to the golden ratio $\varphi = \frac{1+\sqrt{5}}{2}$. Using FFT sizes derived with this approach removed sharp lines from the sensitivity spectrum of the discriminator, and left a pattern closer to noise. Training using these updated settings proved to lack the previous periodic artifacts in the spectrum. The final chosen FFT sizes are $\{78, 126, 206, 334, 542, 876, 1418, 2296\}$.

### B.5.1 LEARNED BIAS DURING TRAINING

We can perform a similar sensitivity analysis on the discriminator during training on single examples from the validation dataset. This allows examination of which parts of a particular sound the discriminator is paying the most attention to. Examining many such plots during late-period training revealed an interesting behavior - the discriminator loss was mainly being influenced by extremely low magnitude parts of the signal spectrum, to the exclusion of the higher energy parts of the spectrum. This behavior would indicate that the discriminator is learning to tell the difference between fake and real by looking at patterns in inaudible parts of the signal spectrum. To attempt to mitigate this behavior, the magnitude of the bins $X_{n,f}$ of the normalized complex spectrograms were scaled by a power law - essentially weighting small magnitude bins lower and higher magnitude bins higher:

$$\hat{X}_{n,f} = X_{n,f} |X_{n,f}|^{\alpha} \tag{13}$$

Experimentally, $\alpha = 1/2$ was found to be an appropriate value. Higher values of $\alpha$ make the discriminator concentrate more on spectral peaks, which can damage overall timbre and intelligibility. A more involved analysis for addressing this issue is left to future work.

## C    APPENDIX: BASELINE AUDIO CODEC MODELS

This section provides a systematic comparison of baseline audio codec models, summarized in Table 12. Characteristics such as causality, training datasets, multilingual speech support, target application domains, and model complexity (measured by parameter count), are described.

| Model | Causal | Training Datasets | Multilingual Speech | Domain | #Params (M) |
|---|---|---|---|---|---|
| Encodec | Optional | DNS, CommonVoice, AudioSet, FSD50K, and Jamendo | Yes | General | 14 |
| DAC | No | DAPS, DNS, CommonVoice, VCTK, MUSDB, and Jamendo | Yes | General | 76 |
| Mimi | Yes | Predominantly English speech (7 million hours) | Likely | Speech | 80 |
| SpeechTokenizer | No | LibriSpeech | No | Speech | 104 |
| SemantiCodec | No | GigaSpeech, multilingual audios from OpenSLR, Million Song Dataset, MedleyDB, MUSDB18, AudioSet, WavCaps, and VGGSound | Yes | General | 507 |

Table 12: Comparison of audio codecs and their characteristics.

## D    APPENDIX: OBJECTIVE METRICS

**Bits Per Second (BPS)** – A metric that reflects the compression efficiency by measuring the number of bits transmitted per second. We'll use this metric to discuss the trade-off between compression and quality.

**Tokens Per Frame (TPF)** – A metric which shows how many parallel tokens are needed for each timestep of the encoded audio. This is important as it effects the easy of modeling the token sequence with a generative model.

**Tokens Per Second (TPS)** – A metric that describes how many tokens are needed per second of audio. This is important as it dictates how much of the context of a generative model is needed per second of encoded audio, if residual tokens are used in flattened form.

**Scale-Invariant Source-to-Distortion Ratio (SI-SDR)** – A waveform-based metric similar to signal-to-noise ratio, with modifications to make it invariant to scale differences (Le Roux et al., 2019). When used alongside spectral metrics, SI-SDR provides insights into the quality of phase reconstruction.

**Mel Distance** – This is a combination of two distances calculated between mel spectrograms of the reconstructed and ground truth waveforms. We use a Hanning window of size $2048$, FFT size of $2048$, hop size of $256$, and $128$ mel bins. The first component is the L1 distance between log-scaled magnitudes. The second component is the spectral convergence calculated between the linear-scaled magnitudes (Steinmetz & Reiss, 2020). Both components are weighted equally.

**STFT Distance** – This utilizes the same two distance measures used in the Mel Distance metric, but with a standard linearly spaced spectrogram. We use a Hanning window of size $2048$, FFT size of $2048$ and hop size of $512$. This metric captures high-frequency fidelity better than the Mel Distance.

**PESQ (Perceptual Evaluation of Speech Quality)** – A speech quality assessment metric that compares the reconstructed speech to a reference, providing a score that correlates with subjective human judgment from $1$ to $5$ (Rix et al., 2001).

**STOI (Short-Time Objective Intelligibility)** – A metric that measures speech intelligibility by comparing short-time spectral envelopes between the reconstructed and ground truth speech. Scores range from $0$ to $1$, where higher values indicate better intelligibility (Andersen et al., 2017).

**MOSNet** – A neural network-based metric that predicts the mean opinion score (MOS) from $1$ to $5$ for speech quality by learning from human-labeled datasets. It offers a reference-free method for estimating perceptual speech quality (Lo et al., 2019).

# E APPENDIX: DEMOGRAPHIC BREAKDOWNS OF THE PERCEPTUAL TEST

The demographic data of participants in the perceptual test reveal that 68.2% of respondents are affiliated with academia, while 31.8% represent industry professionals. A majority (59.1%) are involved in audio or music production and research, highlighting a strong relevance to our listening test. Regarding equipment used during the perceptual test, 63.6% of participants relied on headphones, 22.7% on laptop speakers, and 13.6% on professional-grade speakers.

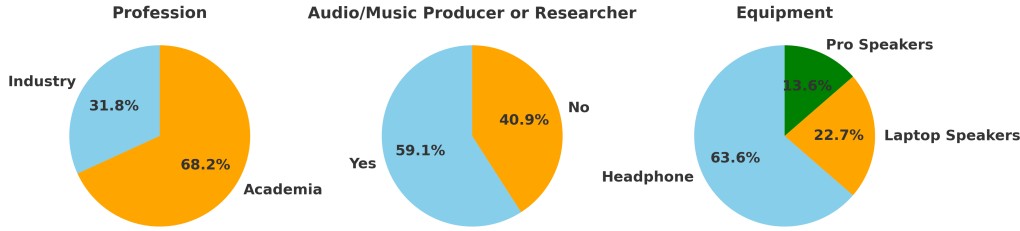

Figure 4: Demographic breakdowns of the perceptual test.

