# OpenReview forum: "Scaling Transformers for Low-Bitrate High-Quality Speech Coding"
_ICLR.cc/2025/Conference — ICLR 2025 Poster_

### Official Review · Reviewer_Baq6 · 2024-11-03

**Soundness:** 3
**Presentation:** 3
**Contribution:** 2
**Rating:** 6
**Confidence:** 4

**Summary:**

The authors present an encoder-decoder transformer-based architecture for encoding speech at very low bitrate, below 1kbps. The Transformer Audio AutoEncoder (TAAE) uses a polyphase time-frequency representation to perform downsampling/upsampling before the enocder and after the decoder of TAAE. Finite Scalar Quantization (FSQ) is employed within the encoder/decoder bottleneck to mitigate codebook underutilization typically seen with vector quantized (VQ) and residual vector quantized (RVQ) approaches. The authors combine an L1 discriminator feature loss with decaying L1 waveform loss and late perceptual reconstruction loss for training. The TAAE is trained on 105k hours of English speech sampled at 16kHz. The reconstruction capability of TAAE is compared to the Descript Audio Codec (DAC), Encodec, SpeechTokenizer, SematiCodec, and Mimi. A mean opinion score (MOS) is also produced from a perceptual evaluation comprised of 23 participants comparing TAAE to Mimi and SemantiCodec. The authors demonstrate that TAAE obtains better reconstruction performance according to both objective measures and MOS. The authors also demonstrate that one variant of the TAAE codebook attains 98% utilization.

**Strengths:**

Originality:
The results presented in Appendix B are enlightening regarding the use of non-power-scaled FFT sizes. The FSQ-based bottleneck also seems to overcome common issues in the training of RVQ systems.

Quality:
The authors provide a wide variety of objective assessments for their architecture's performance. The authors also do a good job of citing current literature.

Clarity:
The authors very clearly describe their architecture and the motivations for their architectural decisions. The appendices were well organized and helpful.

Significance:
Appendix B and the FSQ bottleneck are worthwhile contributions.

**Weaknesses:**

I do not think it is novel to scale the parameter count of a neural network autoencoder and demonstrate better compression ratios/reconstruction compared to smaller architectures.  This is a well known result.

I do not think it is novel to restrict the domain of a neural network autoencoder in comparison to another architecture trained on a more general domain and demonstrate better compression ratios/reconstruction. This is a well known result.

By restricting the domain of their speech audio corpus to English speech, the authors have produced an English speech audio codec. In order to claim that this is a "speech codec," the authors should evaluate on non-English speech to demonstrate generalization capabilities.

I do not think DAC, Encodec, and SemantiCodec are reasonable baselines to compare to, as none claim to be English speech codecs.

Mimi focuses on streaming and causality with 1/10 the parameter count of TAAE, which makes no claims regarding streaming capability. This also leads to an odd comparison as the goals of Mimi and TAAE are not aligned.

It is unclear why SpechTokenizer was left out of the perceptual evaluation, as it is the most comparable to TAAE in terms of architecture and training domain. Comparison to SpeechTokenizer could also boost claims that the FSQ significantly outperforms RVQ schema.

I think the presented MOS scores are confusing, as MOSNet has estimated the MOS closer to 3 than to 5. A MUSHRA evaluation should have been used instead for pairwise comparison between codecs and is standard in the literature cited in this paper. Furthermore, the authors should include demographic breakdowns of the perceptual evaluation, as well as a description of the listening setup, as is standard for speech codecs.

**Questions:**

Why was SpeechTokenizer not included in the perceptual evaluation?

How do the design goals of Mimi align with that of TAAE? Why is Mimi a good baseline comparison to TAAE?

Why was a MOS evaluation chosen instead of MUSHRA?

How do you explain the gap in your perceptual evaluation MOS score and the estimate provided by MOSNet?

Who was included in the perceptual evaluation, and what was their listening setup?

How does TAAE perform on non-English speech? And how does that compare to the more generalist NAC?

---

> ### Author Response · Authors · 2024-11-22
> **Response to reviewer Baq6 (Part 1/3)**
>
> Dear reviewer Baq6,
>
> Thank you for your efforts in reviewing this paper, and for raising your concerns with the work. We have grouped your questions and comments and provided detailed responses to each point below.
>
> > Q1: I do not think it is novel to scale the parameter count of a neural network autoencoder and demonstrate better compression ratios/reconstruction compared to smaller architectures. This is a well known result.
>
> We agree that scaling is a well known result in other fields (e.g. LLMs), which is one of the primary motivations for us pursuing this direction (as articulated in **Sec.1**). However, we’re not aware of existing work showing that **audio codec** performance scales with parameter count. If we’re missing some important work that establishes this, it’d be much appreciated if you could share it. Our opinion (explained in **Sec.1**) is that this was not a goal for previous codec models as they were more directly targeting the transmission/storage use-cases, which benefit much more from small parameter-efficient models. This is also likely to be related to the overall difficulty of scaling CNNs to the billion parameter level (see e.g. ConvNeXTv2). In our work we establish that:
> 1. Leveraging transformer blocks as the primary component of a model is viable for an end-to-end audio waveform codec, once a number of architectural challenges are overcome.
> 2. Scaling parameter count does indeed provide improved performance in audio waveform coding (which we also now demonstrate more directly in the new ablation in **Appendix A.2**).
>
> We strongly believe that both are novel contributions to the field of audio coding, which will be helpful in informing future work.
>
> > Q2: I do not think it is novel to restrict the domain of a neural network autoencoder in comparison to another architecture trained on a more general domain and demonstrate better compression ratios/reconstruction. This is a well known result. I do not think DAC, Encodec, and SemantiCodec are reasonable baselines to compare to, as none claim to be English speech codecs.
>
> We are certainly not claiming this as a point of novelty, nor is it something we are trying to demonstrate.  We acknowledge the relatively limited domain (English speech) as a limitation of this initial iteration of our model, and we intend to address this by scaling to more diverse datasets in the future. This has been made more explicit in **Sec.3.7**. We would argue that the presented results, along with the new experiment in **Appendix A.5** which shows good generalization to unseen languages, strongly suggest that model performance will increase even further when extended to a larger and more diverse dataset. Our opinion is that this enhances the attractiveness of this architecture as the basis for future work.
>
> Regarding the fairness of baselines choices, we agree that all the chosen baselines were not exactly designed with the same goals as our proposed model, nor necessarily trained on the same data. This is not an intentional choice on our part, as we’re restricted by which models are released to the public. The chosen baselines represent generally the most widely used and positively considered neural codecs for speech, showing strong performance. Hopefully there will be fairer baselines available in the future for this comparison, as larger neural audio codecs specifically targeting downstream generative use cases become more prevalent.

---

> > ### Author Response · Authors · 2024-11-22
> > **Response to reviewer Baq6 (Part 2/3)**
> >
> > >Q3: By restricting the domain of their speech audio corpus to English speech, the authors have produced an English speech audio codec. In order to claim that this is a "speech codec," the authors should evaluate on non-English speech to demonstrate generalization capabilities. How does TAAE perform on non-English speech? And how does that compare to the more generalist NAC?
> >
> > We’ve conducted an extensive set of generalization tests using Multilingual LibriSpeech, with results presented in **Appendix A.6**. The performance remains stable, showing no significant degradation when processing unseen languages. The performance of TAAE is either better than or comparable to models trained on multiple languages. This finding highlights TAAE’s potential for even greater performance when trained on multilingual data. We appreciate your suggestion to explore this further. Additionally, multilingual audio examples have been updated on the anonymous website.
> >
> > >Q4: Mimi focuses on streaming and causality with 1/10 the parameter count of TAAE, which makes no claims regarding streaming capability. This also leads to an odd comparison as the goals of Mimi and TAAE are not aligned. How do the design goals of Mimi align with that of TAAE? Why is Mimi a good baseline comparison to TAAE?
> >
> > The main reason for choosing Mimi as a baseline is simply that it is the model which previously demonstrated the best reconstruction performance at very low bit-rates, outperforming other non-streaming speech/audio codecs in this range. We agree that the design goals are not aligned, but as stated before we are restricted by what models are available publicly. It is also an interesting point of comparison as it allows us to contrast a fairly traditional CNN-based codec with transformer blocks in the bottleneck with an architecture that relies mainly on patching and transformers.
> >
> > In order to make this comparison more meaningful we added a new experiment, described in **Appendix A. 4**. In this experiment we finetuned the main presented version of TAAE to be fully causal, using causal convolution and causal attention windows. The results show that this causal TAAE is marginally degraded compared to our non-causal version, and outperforms Mimi in some objective metrics, despite being trained with significantly fewer steps and data hours. Additionally, audio examples generated by the causal TAAE model have been updated on the anonymous website.
> >
> >
> > | **Model**            | **BPS** | **SI-SDR ↑** | **Mel ↓** | **STFT ↓** | **PESQ ↑** | **STOI ↑** | **MOSNet ↑** |
> > |-----------------------|---------|--------------|-----------|------------|------------|------------|--------------|
> > | Mimi                 | 1100    | 2.20         | 0.94      | 1.31       | 3.01       | 0.90       | 3.24         |
> > | TAAE (causal)        | 700     | 4.04         | 0.94      | 1.31       | 3.09       | 0.92       | 3.34         |
> > | TAAE (non-causal)    | 700     | 4.73         | 0.86      | 1.26       | 3.09       | 0.92       | 3.36         |

---

> > > ### Author Response · Authors · 2024-11-22
> > > **Response to reviewer Baq6 (Part 3/3)**
> > >
> > > > Q5: It is unclear why SpeechTokenizer was left out of the perceptual evaluation, as it is the most comparable to TAAE in terms of architecture and training domain. Comparison to SpeechTokenizer could also boost claims that the FSQ significantly outperforms RVQ schema. Why was SpeechTokenizer not included in the perceptual evaluation?
> > >
> > > Thank you for your suggestion. We agree that including SpeechTokenizer in the perceptual evaluation is important. We have now incorporated the SpeechTokenizer (1500 bps) variant into our new subjective test. The results show that SpeechTokenizer (1500 bps) outperforms SemantiCodec (340 and 680 bps) and Mimi (550 bps) but performs worse than Mimi (1100 bps) and TAAE (400 bps and 700 bps). The updated results are presented in Figure 2.
> > >
> > > > Q6: I think the presented MOS scores are confusing, as MOSNet has estimated the MOS closer to 3 than to 5. How do you explain the gap in your perceptual evaluation MOS score and the estimate provided by MOSNet?
> > >
> > > Thank you for your observation regarding the MOS scores. To clarify, MOSNet is a neural network model trained to correlate with human MOS ratings, but its estimates are influenced by the quality and characteristics of its training data - which may introduce some inaccuracies or biases. Furthermore, in our previous MOS test, the goal was to evaluate how the reconstructed audio sounds compared to the ground truth. Given that the test data's quality may not be exceptionally high, there is naturally some difference between the MOS scores and the estimates provided by MOSNet. The human MOS scores in our previous test reflect subjective evaluations based on the perceived quality of the reconstructions relative to the ground truth, while MOSNet estimates are based on its learned correlations regarding speech quality. Our evaluated MOSNet scores for the baselines are consistent with the results reported in the Moshi (Mimi) technical report.
> > >
> > > > Q7: A MUSHRA evaluation should have been used instead for pairwise comparison between codecs and is standard in the literature cited in this paper. Why was a MOS evaluation chosen instead of MUSHRA? Furthermore, the authors should include demographic breakdowns of the perceptual evaluation, as well as a description of the listening setup, as is standard for speech codecs. Who was included in the perceptual evaluation, and what was their listening setup?
> > >
> > > We re-ran the perceptual tests using a MUSHRA setup and included SpeechTokenizer as an additional baseline. While the results are not significantly different from our previous MOS evaluation, we agree that the MUSHRA format is more appropriate, as it aligns better with standard references in the literature. Additionally, we provided further details about the listening setup in Section 3.3 and included demographic breakdowns of the MUSHRA evaluation in Appendix E.
> > >
> > > | Model                     | MUSHRA Score |
> > > |---------------------------|--------------|
> > > | SemantiCodec (340 bps)    | 51.06        |
> > > | Mimi (550 bps)            | 57.48        |
> > > | SemantiCodec (680 bps)    | 59.13        |
> > > | SpeechTokenizer (1500 bps)| 70.67        |
> > > | Mimi (1100 bps)           | 78.62        |
> > > | TAAE (400 bps)            | 88.17        |
> > > | TAAE (700 bps)            | 89.50        |
> > > | Ground Truth              | 92.20        |
> > >
> > > Thank you for your many insightful comments and suggestions. We hope you agree that the paper is much improved with these additions. If you have further topics for us to address, please share and we will be happy to tackle them.

---

> ### Author Response · Authors · 2024-11-26
> **Reminder to check our changes**
>
> Hi reviewer Baq6,
>
> We'd really appreciate if you could take a look at the changes we made to address you're concerns. We think the results on multi-lingual generalisation are especially interesting! We also made some significant changes to address your other concerns. We're available here to discuss.

---

> > ### Comment · Reviewer_Baq6 · 2024-11-28
> >
> > Thank you for your in-depth response. I very much appreciate the extra work you've done to address my concerns. The multilingual evaluation is particularly illuminating.
> >
> > >However, we’re not aware of existing work showing that audio codec performance scales with parameter count. If we’re missing some important work that establishes this, it’d be much appreciated if you could share it.
> > A quick scan of "High-Fidelity Audio Compression with Improved RVQGAN" by Kumar et al shows the following sentence at the beginning of Section 4.5 Ablation Study: "We find that varying the decoder dimension has some effect on performance, with smaller models having consistently worse metrics."
> >
> > A neural network audio codec is still a neural network, and is governed by the same principles as any other neural network.
> >
> >
> > >To clarify, MOSNet is a neural network model trained to correlate with human MOS ratings, but its estimates are influenced by the quality and characteristics of its training data - which may introduce some inaccuracies or biases.
> > Given these well known issues with MOSNet, it is worth considering how including this metric bolsters your arguments. Uncharitable readings of the presented evaluation could suggest examples were cherry-picked for the subjective listening test and that overall performance is lacking. Regardless, the rest of the objective evaluation makes a convincing case that TAAE outperforms the selected methods.
> >
> >
> > >>We re-ran the perceptual tests using a MUSHRA setup and included SpeechTokenizer as an additional baseline.
> > Thank you for addressing this. For future reference, MUSHRA tests are often followed by a significance test, such as a Post hoc Tukey HSD ANOVA, to observe whether ratings significantly differ from one another, or if differences are the result of random chance. Some type of error bar or standard deviation would help readers quickly glean the performance of your algorithm compared to others.
> >
> > In light of your follow-up evaluation, I have raised my rating of the paper.

---

> > > ### Author Response · Authors · 2024-11-29
> > > **Thanks to reviewer Baq628**
> > >
> > > Hi reviewer Baq6,
> > >
> > > Thank you for reviewing our changes and increasing your rating. We agree with you about the shortcomings of MOSNet. It was included primarily due to it's inclusion in recent related works (e.g. the Moshi/Mimi paper).
> > >
> > > Thank you again for your work on reviewing this paper. We are glad that you raised the objections that you did, as addressing them has made this a stronger paper.

---

### Official Review · Reviewer_bLHu · 2024-11-03

**Soundness:** 3
**Presentation:** 2
**Contribution:** 3
**Rating:** 8
**Confidence:** 5

**Summary:**

This paper proposed  TAAE, an audio codec model that uses Transformer encoders as the main building block to replace conventional convolution-based modules. To accommodate the choice, TAAE performs downsampling mainly by patchifying the time domain signal and training transformer encoder stacks on top of the downsampled sequence. For discretizing audio, TAAE relied on FSQ-based bottleneck that approximates continuous low-dimensional latent numerically. Experiment results show TAAE achieved outstanding speech quality on autoencoding at a significantly lower bit rate comparing to existing models.

**Strengths:**

- The idea of using Transformer and the main architecture for the neural audio codec learning is novel and well executed.
- Judging from the audio samples on the demo page and MOS study, TAAE is clearly state-of-the-art in low bit rate speech compression.
- This paper provided a lot of detailed knowledge, empirical findings, and engineering improvements that can truly benefit the audio codec research community. I personally learned a lot in the details such as the discussion on systematic bias of discriminator, choice of filterbank, observation on the latent representation of silence frames with self-attention, etc.

---

### Justification for rating
Overall, I believe this work is novel enough and provides solid contributions to the field.
However, some improvement might be necessary (see weaknesses below).
If the authors can properly address these concerns and update the submission accordingly, I would be more than happy to raise my rating on this paper.

**Weaknesses:**

Given the main contribution of this work is in exploring an alternative architecture for codec models, completeness in terms of design details and reproducibility are expected. In contrast, I found a lot of details missing or vague. (Although the authors state the code will be released later, the paper itself should still be comprehensive alone.) Here are some examples:

---

> ($\S$2.1) ... Instead we raise the $\epsilon$ constant used in the calculation of normalization factors in the layer norm blocks ... allows convergence of the architecture.

This appears to be an interesting observation and a critical hyper-parameter for training the model as the authors spent a paragraph discussing it, but neither the exact value nor the study/experiment on $\epsilon$  is provided.

---

> ($\S$2.4)...For training the codec itself, we primarily use a normalized feature-matching L1 loss on the per-layer features of the discriminator network ... In addition we found it beneficial to include a traditional L1 reconstruction loss to boost convergence at the beginning of the training process ...

The overall objective of the model is not explicitly given but described in a more hand-wavy style instead, which could easily lead to misunderstanding. The full objective should be listed explicitly together with the weight/importance for each term/component in the main paper or appendix.

---

> ($\S$2.1) ... The self-attention uses a sliding window of size 128, to restrict receptive field and aid generalization of the architecture to arbitrary length sequences.

This choice appears as one simple sentence, but self-attention is the key difference between TAAE and prior works, which changes the properties of the model dramatically.
If my understanding is correct, this means the receptive field of the first layer is already 2.56 seconds (128 frames $\times$ 20 ms-per-frame), and the number doubles for every layer. It is obvious that TAAE has a much larger receptive field size comparing to convolution-based models. While this is an advantage, it could also lead to some problems that are not discussed in the paper.

- What is the trade-off between length generalization and sliding window size for TAAE? How do time complexity and empirical inference time change accordingly? How do these numbers compare to those of CNN-based models?
- Beyond length generalization, can TAAE perform streaming encoding/decoding (as most of the existing works compared in this paper)?
  - If so, what is the size receptive field? how does it affect the latency of the codec? how does it compares to conventional CNN-based codec models?
  - If not, this should still be explicitly discussed as a limitation of the proposed framework in the paper.

These are just some examples. In short, I believe the fundamental differences between TAAE and CNN-based codec models should be discussed in the paper more throughout and carefully. Both advantages and disadvantages should be clearly stated and summarized in the main body of the paper.

---

I believe these concerns can all be addressed without additional training, thus should be easy enough to complete within the rebuttal period.

**Questions:**

(please see weaknesses above)

---

> ### Author Response · Authors · 2024-11-22
> **Response to reviewer bLHu (Part 1/2)**
>
> Dear reviewer bLHu,
>
> Thank you for your efforts in reviewing this paper, and for your many insightful comments. We’ve tried to address your concerns with the following changes:
>
> >Q1: This appears to be an interesting observation and a critical hyper-parameter for training the model as the authors spent a paragraph discussing it, but neither the exact value nor the study/experiment on ϵ is provided.
>
> In **Appendix B.1** we have added some further discussion about the ϵ value of the layer norms, as well as the exact value used in the experiments. Thank you for spotting this missing piece of information.
>
> >Q2: The overall objective of the model is not explicitly given but described in a more hand-wavy style instead, which could easily lead to misunderstanding. The full objective should be listed explicitly together with the weight/importance for each term/component in the main paper or appendix.
>
> We have expanded **Sec. 2.4**. to give much more detail about the training objective, in both the pretraining and finetuning stages.
>
> >Q3: What is the trade-off between length generalization and sliding window size for TAAE? How do time complexity and empirical inference time change accordingly? How do these numbers compare to those of CNN-based models?
>
> * We’ve added **Appendix B.2** discussing the relative receptive fields of our proposed model and the baselines. Interestingly, our proposed model does not have a wildly different receptive field to existing models - partly as very few are purely convolutional. Several baselines use RNNs with effectively unlimited receptive fields, and use chunked inference to counteract any downside from this.
>
> * We’ve added an experiment addressing length generalization by testing inference of the TAAE and baselines with a variety of utterances lengths. This can be found in **Appendix A.6**. The results show mild degradation at longer utterances, which can be mitigated if necessary by chunked inference.
>
> * We’ve added an experiment which examines empirical inference time. This can be found in **Appendix A.9**. This experiment shows that the TAAE architecture is competitive with the baselines in terms of empirical inference time, despite utilizing a much larger number of parameters. The architecture benefits greatly from the patched representation taking care of much of the down/upsampling, as well as the large amount of effort invested by other researchers and engineers in optimizing the components of the transformer architecture.
>
> * We have interpreted ‘time-complexity’ to mean Big O analysis with respect to varying sequence length. This is now discussed briefly in **Appendix B.2**. Our model (and all the baselines) scale as O(n) with sequence length.

---

> > ### Author Response · Authors · 2024-11-22
> > **Response to reviewer bLHu (Part 2/2)**
> >
> > >Q4: Beyond length generalization, can TAAE perform streaming encoding/decoding (as most of the existing works compared in this paper)? If so, what is the size receptive field? how does it affect the latency of the codec? how does it compares to conventional CNN-based codec models?
> >
> > We trained a causal version of the codec as a finetune from the main presented model. This is presented in **Appendix A.4**. We show objective metrics vs our main model and vs Mimi, which show only minor degradation in causal mode. We’ve also included discussion of latency in **Appendix B.2**. The anonymous website has been updated with new audio examples generated by the causal TAAE model.
> >
> > | **Model**            | **BPS** | **SI-SDR ↑** | **Mel ↓** | **STFT ↓** | **PESQ ↑** | **STOI ↑** | **MOSNet ↑** |
> > |-----------------------|---------|--------------|-----------|------------|------------|------------|--------------|
> > | Mimi                 | 1100    | 2.20         | 0.94      | 1.31       | 3.01       | 0.90       | 3.24         |
> > | TAAE (causal)        | 700     | 4.04         | 0.94      | 1.31       | 3.09       | 0.92       | 3.34         |
> > | TAAE (non-causal)    | 700     | 4.73         | 0.86      | 1.26       | 3.09       | 0.92       | 3.36         |
> >
> > >Q5: I believe the fundamental differences between TAAE and CNN-based codec models should be discussed in the paper more throughout and carefully. Both advantages and disadvantages should be clearly stated and summarized in the main body of the paper.
> >
> > We’ve added much more context and discussion about CNNs vs transformers, including some broader discussion of rationale in **Sec.1** and **Sec.2.1** within the main text and some more detailed explanation behind our motivations in **Appendix B.3**. Our main motivation for exploring a transformer-based architecture is the scaling properties shown in other domains, but we also believe that attention may fundamentally be better placed to address the irregular distribution of information in audio and speech waveforms.
> >
> > Thank you again for your extremely helpful perspectives and feedback. If you have any more comments or concerns, we’re happy to address them.

---

> > > ### Comment · Reviewer_bLHu · 2024-11-24
> > >
> > > I would like to thank the authors for addressing my concerns. With the additional implementation details and in-depth analysis, I believe this work can make a significant contribution to the field. I have raised my rating accordingly.

---

### Official Review · Reviewer_VKr9 · 2024-11-04

**Soundness:** 4
**Presentation:** 3
**Contribution:** 3
**Rating:** 8
**Confidence:** 5

**Summary:**

This work describes an approach to leverage scaled transformer model architecture to achieve low-bitrate and high-quality speech coding. Different from conventional neural audio codec approaches, this work leverages the finite scalar quantization (FSQ) based bottleneck to achieve high codebook utilization rate and avoid the difficulty in training a VQ-VAE auto-encoder. Experimental results show this works outperform existing baselines in both objective and subject tests.

**Strengths:**

This paper is well written and very clear to follow. In the introduction part, it clearly presents the motivations and has an excellent survey of the existing methods.

Though using transformers to scale and leverage FSQ for high codebook utilization is not something new, this paper presents the motivations of these changes, the associated challenges and their mitigations. This paper also introduces a new method so that FSQ can be used in a similar way as RVQ where a varying bits-per-second rate can be achieved.

This paper presents strong experiment results, significant improving over the existing baselines (but at the cost of increased computation and latency).

**Weaknesses:**

If I understand the proposed model correctly, it is based on transformer layer with a local attention of 128 (both left and right), which means different from DAC/Encodec/Mimi etc which use causal encoders, the encoder in the proposed method is not causal, and it will introduce a latency up to the patch length (which is 320/16k ~ 20ms?). It would be great if the author can present the results with causal encoder so that it can be compared with DAC/Encodec/Mimi in a relative fair comparison (apart from the model size difference).

**Questions:**

N/A

---

> ### Author Response · Authors · 2024-11-22
> **Response to reviewer VKr9**
>
> Dear reviewer VKr9 ,
>
> Thank you for reading our paper and for providing such insightful comments. We made the following changes to address your review:
>
> >Q1: “It would be great if the author can present the results with causal encoder so that it can be compared with DAC/Encodec/Mimi in a relative fair comparison (apart from the model size difference).”
>
> * We trained a causal version of the codec as a finetune from the main presented model. This is presented in **Appendix A.4**. We achieved this by switching convolutions to causal versions and changing the sliding attention window to be purely causal. The model performs very close to the main model presented in the paper, so the trade-off for applying the model in a streaming situation appears to be minimal, assuming sufficient computation resources are available to execute the model. We did not want to alter the main intention of the paper (we were not concentrated on streaming use-cases), so this is kept as an extension rather than used to replace the non-causal version of the model in the main comparisons. New audio examples created by the causal TAAE model can be found on the anonymous website.
>
> | **Model**            | **BPS** | **SI-SDR ↑** | **Mel ↓** | **STFT ↓** | **PESQ ↑** | **STOI ↑** | **MOSNet ↑** |
> |-----------------------|---------|--------------|-----------|------------|------------|------------|--------------|
> | Mimi                 | 1100    | 2.20         | 0.94      | 1.31       | 3.01       | 0.90       | 3.24         |
> | TAAE (causal)        | 700     | 4.04         | 0.94      | 1.31       | 3.09       | 0.92       | 3.34         |
> | TAAE (non-causal)    | 700     | 4.73         | 0.86      | 1.26       | 3.09       | 0.92       | 3.36         |
>
> * We have added some discussion in **Appendix B.2** which compares the receptive fields, causality and latency of our proposed model and the baselines.
>
> Thank you again for your reviewing efforts. We’re happy to address any more comments or concerns you may have.

---

### Official Review · Reviewer_1mKc · 2024-11-04

**Soundness:** 3
**Presentation:** 2
**Contribution:** 3
**Rating:** 6
**Confidence:** 4

**Summary:**

The paper presents a new Transformer architecture for speech coding. It is characterized by the new scalar quantization method performed in a dimension-reduced space, which showed improved coding gain compared to other methods that are based on residual vector quantization. The paper also provides various training strategies that appear to be useful for neural codec training.

**Strengths:**

- The proposed system appears to work well according to the objective metrics and subjective tests.
- The proposed FSQ idea seems to be a solid quantization option, improving the codebook utilization.
- The authors put a lot of effort in making it more scalable by adding multiple levels of quantization.

**Weaknesses:**

- The proposed method relies on the dimension reduction part for its dimension-specific scalar quantization to work. And that's why they could achieve higher codebook utilization. Meanwhile, there is also a trend that higher codebook utilization leads to lower coding gain if entropy coding is applied after tokenization. Indeed, the paper does not mention anything about Huffman coding results, which the proposed method might not be able to take advantage of due to the low dimensionality and high codebook utilization. At the same time, the RVQ-based ones might have a better chance of compressing more via Huffman coding. I wish the paper provided an in-depth discussion about it. In my opinion, all the coding gain and performance-related arguments must be based on the entropy-coded bitrates of all codecs mentioned.

- The other main criticism is that the proposed model is just a lot bigger than the other models. I don't mean to argue that a bigger codec necessarily results in a better coding gain, but in general, it is also true that there is a relation. I wish the paper had provided an ablation test that investigated the impact of the different sizes of the proposed model.

- The paper provides various tips and useful information about their model training, but they are scattered in different places without a clear organization.

**Questions:**

- How does it compare to the HuBERT-based codec? HuBERT can be as large as the proposed model (its X-Large version), and can turn into a codec with a vocoder attached to it as shown in [a].

[a] A. Polyak et al. “Speech Resynthesis from Discrete Disentangled Self-Supervised Representations,” Interspeech 2021

---

> ### Author Response · Authors · 2024-11-22
> **Response to reviewer 1mKc (Part 1/2)**
>
> Dear reviewer 1mKc,
>
> Thank you for your efforts in reviewing this paper, and for the very insightful comments. We’ve made a number of modifications in order to address your concerns.
>
> >Q1: "The proposed method relies on the dimension reduction part for its dimension-specific scalar quantization to work. And that's why they could achieve higher codebook utilization. Meanwhile, there is also a trend that higher codebook utilization leads to lower coding gain if entropy coding is applied after tokenization. Indeed, the paper does not mention anything about Huffman coding results, which the proposed method might not be able to take advantage of due to the low dimensionality and high codebook utilization. At the same time, the RVQ-based ones might have a better chance of compressing more via Huffman coding. I wish the paper provided an in-depth discussion about it. In my opinion, all the coding gain and performance-related arguments must be based on the entropy-coded bitrates of all codecs mentioned."
>
> Our initial draft did not contain discussions on the Huffman coded bitrates of the proposed model or the baselines, as our use-case for this model was not focused on transmission or storage of speech. However, we agree that this information is important and valuable to the wider speech coding community. To address this, we have added a new section in **Appendix A.8** that examines and computes the codebook utilisation and Huffman-coded bitrates for the proposed model and the baselines. Our analysis shows that Huffman coding achieves a moderate reduction in bits-per-second for earlier RVQ-based codecs such as Encodec. In contrast, more recent RVQ-based codecs (e.g., DAC, SpeechTokenizer, Mimi), which leverage advanced methods like EMA updates, factorized codes, and L2-normalized embeddings to boost RVQ codebook utilisation significantly, derive only marginal benefits from Huffman coding. Similarly, FSQ-based tokens produced by TAAE achieve comparable levels of gains in coding efficiency with Huffman coding to these modern RVQ models.
>
> >Q2: "The other main criticism is that the proposed model is just a lot bigger than the other models. I don't mean to argue that a bigger codec necessarily results in a better coding gain, but in general, it is also true that there is a relation. I wish the paper had provided an ablation test that investigated the impact of the different sizes of the proposed model."
>
> We conducted scaling experiments with TAAE architectures containing approximately 250M, 500M, and 1B parameters, which is shown in **Appendix A.2**. The results demonstrate that scaling up the parameter count yields clear improvements in objective metrics, although the smaller models still perform respectably compared to the baselines.
>
> | Param. Count | SI-SDR ↑ | Mel ↓ | STFT ↓ | PESQ ↑ | STOI ↑ |
> |--------------|----------|-------|--------|--------|---------|
> | 240M         | 3.52     | 1.24  | 1.67   | 2.74   | 0.87    |
> | 540M         | 4.31     | 1.21  | 1.66   | 2.80   | 0.88    |
> | 950M         | 4.80     | 1.18  | 1.59   | 2.82   | 0.88    |
>
> We would also like to highlight that scaling transformer-based codec architectures to 1B parameters is a key contribution of this work. Unlike traditional CNN-based codecs, TAAE uses a transformer-based architecture that offers enhanced scalability. However, in our early experiments we found it challenging to make Transformers work effectively for audio coding tasks. The success of the TAAE model is due to our specific architecture design, empirical findings, and optimizations, as described in **Section 2** and further analyzed in **Appendix B**.

---

> > ### Author Response · Authors · 2024-11-22
> > **Response to reviewer 1mKc (Part 2/2)**
> >
> > >Q3: "HuBERT can be as large as the proposed model (its X-Large version), and can turn into a codec with a vocoder."
> >
> > We included an investigation of a HuBERT + HiFi-GAN Vocoder model as a baseline (unitHiFi-GAN). However, we were unable to find pretrained vocoder models compatible with HuBERT-XL and had to rely on a publicly available release using the smaller HuBERT-base (95M). The results of this investigation are presented in **Appendix A.7**.
> >
> > We did not initially consider this as a baseline, as it is arguably a conditional generative model rather than a codec, and should be evaluated from a different perspective. The HuBERT-based codec approach benefits from its semantic representations learned through self-supervised objectives. However, it may introduce trade-offs, such as the loss of acoustic details and timbre in the resynthesized audio and would suffer greatly in objective metrics, which was confirmed by our results. Additionally, we updated the unitHiFi-GAN reconstructed speech examples on our anonymous website for reference. While the unitHiFi-GAN model generates plausible speech, it struggles to preserve speaker identity.
> >
> > These limitations could potentially be addressed by incorporating discrete tokens from additional speech models (e.g., for pitch tracking or speaker classification), and the HuBERT-XL (1B) model might outperform the HuBERT-BASE (95M) model in this context. However, our work focuses on developing an end-to-end waveform codec model designed to achieve high-quality reconstruction while maintaining low-bitrate compression, distinguishing our approach from HuBERT-based methods.
> >
> > | Model         | BPS  | SI-SDR ↑  | Mel ↓  | STFT ↓ | PESQ ↑ | STOI ↑ | MOSNet ↑ |
> > |---------------|------|-----------|--------|--------|--------|--------|----------|
> > | unitHiFi-GAN  | 500  | -45.95    | 3.14   | 3.24   | 1.12   | 0.16   | 2.98     |
> > | TAAE          | 400  | 3.18      | 0.97   | 1.35   | 2.96   | 0.90   | 3.36     |
> > | TAAE          | 700  | 4.73      | 0.86   | 1.26   | 3.09   | 0.92   | 3.36     |
> >
> >
> > >Q4: "The paper provides various tips and useful information about their model training, but they are scattered in different places without a clear organization."
> >
> > We have done some re-organization of the paper to hopefully make the presentation more consistent. The ‘Architecture’ section now only contains high-level discussion of the architecture, with more detailed discussion moved to the Appendices and quantitative information moved to the ‘Experiments’ section. We have also added a number of paragraphs allowing readers to understand which additional experiments were performed, and where to find them.
> >
> > Thanks again for your efforts, please let us know if there are any other changes that would improve the paper in your view.

---

> ### Author Response · Authors · 2024-11-26
> **Any further concerns?**
>
> Hi reviewer 1mKc,
> Thanks for updating your score! If you have any remaining concerns with the work, please feel free to share and we can discuss.

---

### Author Response · Authors · 2024-11-22
**General comments to all reviewers**

Dear reviewers,

We have now uploaded a new version of the paper with a significant amount of new content that we believe should address the majority of your comments. We've replied to each of you individually to give specific replies to your concerns, but we also wanted to highlight here the new content suggested by reviewers.

* We performed an ablation showing that the performance of the TAAE scales clearly with parameter count. This is described in **Appendix A.2**.

* We trained a causal finetune of the presented model. This demonstrates minor performance degradation compared to the main presented model but is competitive with the strongest streaming-focused baseline. This is described in **Appendix A.4**.

* We re-ran the subjective perceptual test with a more conventional MUSHRA format, and the addition of SpeechTokenizer as an extra baseline. The results are consistent with those presented in the previous version of the paper.

* We tested the generalization of the presented model and baselines to unseen languages, by evaluating objective metrics on the Multilingual LibriSpeech dataset. The model shows strong generalization performance, reaching parity or outperforming models trained on multi-lingual or general audio data. This is presented in **Appendix A.5**.

* We tested the generalization of the model and baselines to utterances of a variety of lengths. This is presented in **Appendix A.6**

* We performed a comparison with an alternative style of codec model, constructed using a pretrained HuBERT semantic model a HifiGAN-derived vocoder model. This is presented in **Appendix A.7**.

* We performed entropy coding experiments on the model and baselines. This revealed that our model and most baselines exhibited high codebook utilization and, and did not benefit strongly from entropy coding. This is presented in **Appendix A.8**.

* We measured RTF for the proposed model and baselines, showing that despite a much larger parameter count, inference time for the TAAE is competitive with baselines. This is presented in **Appendix A.9**

* We added further clarification about the training objective in **Sec.2.4**, and clarified many hyperparameters in **Sec.3.2**

* We added discussion of the receptive field, causality and latency of the presented model and baselines as well as additional discussion about the merits of convolution vs attention. This is presented in **Appendix B2--B3**

* We added new audio examples demonstrating the generalization of the model to unseen languages and also demonstrating the causal variant of the model. These can be found at the following link: https://taae-iclr-2025.github.io/taae_anonymised/

Thank you for working with us on making this paper the best it can be! We very much appreciate your input.

---

### Public Comment · ~Zhen_Ye2 · 2024-11-28
**Excellent paper**

Hello,

I am very, very impressed with your work, as I am currently researching codecs with single VQ and Transformer decoders. Your work has given me a lot of inspiration. If it's convenient, I would love to discuss it with you.

First, I tried your proposed new discriminator  FFT parameters {78, 126, 206, 334, 542, 876, 1418, 2296} to replace old one { 128, 256, 512, 1024, 2048} and found it works exceptionally well, improving the speaker similarity before and after from 0.80 to 0.82.

My current experiments are trained on 16kHz audio, using 50 tokens per second (50x1), with an STOI of about 0.91 and a PESQ of 3.01 and sim of 0.82. My decoder is a Transformer (hidden size 1024, 12 layers) with vocos. I'm sorry—I really wanted to try your Patched Transform scheme, but I'm not familier with DSP, so implementing the code is hard for me. I'm also using FSQ with a codebook size of 65536 (4^8).

I have a few questions I'd like to ask you:

1. Have you tried the 50Hz x1 setting? Or, what performance do you expect your method to achieve under this setting?

2. I noticed that the number of channels in your discriminator implementation is 256. I'm using the MRD from DAC ([link](https://github.com/descriptinc/descript-audio-codec/blob/c7cfc5d2647e26471dc394f95846a0830e7bec34/dac/model/discriminator.py#L136)). I tried setting the channels to 256, and the discriminator's parameters are about 100M. Is this correct? How does it compare to the number of parameters in your discriminator?

3. Regarding SYSTEMATIC BIAS, I tried it but suspect there may be an issue with my implementation. Here's my code; I'm not sure if I understood your paper correctly:

```python
x_stft = torch.stft(x, fft_size, hop_size, win_length, window.to(x.device), return_complex=True)
gamma = 1/2
X_magnitude = torch.abs(x_stft)
x_stft = x_stft * torch.pow(X_magnitude + 1e-8, gamma)
```

4. About the encoder, I tried using mel + Transformer or STFT + Transformer, but the former resulted in worse performance, and the latter's performance was similar to DAC's encoder (about 40M parameters). I'm not sure if your method has experimented with different encoders. If you still use the CNN-based DAC encoder, is there a significant difference in your method? My thinking about the encoder is that since the encoder's features have to go through VQ, it means that no matter how powerful your encoder is, after VQ, it can only retain key information. Does this mean that for the encoder, maybe we don't need a very powerful model—perhaps DAC's encoder is sufficient?

If you are busy, please don't feel pressured to reply immediately, but I am really looking forward to discussing this with you.

Thank you very much for your paper. I have discussed your paper with many collaborators working on codec, and we all think it's an excellent paper. Your discussions on various technical details are really great. I'm very certain that your work will have a significant impact on the development of audio codec!

Looking forward to your reply!

---

> ### Author Response · Authors · 2024-11-28
> **Answers to Zhen's questions**
>
> Hi Zhen,
>
> Thank you for your kind words. We're happy that you enjoyed the paper!
>
> >I'm sorry—I really wanted to try your Patched Transform scheme, but I'm not familiar with DSP, so implementing the code is hard for me.
>
> Here's an implementation for you:
> ```
> class PatchedPretransform(nn.Module):
>     def __init__(self, channels, patch_size):
>         super().__init__()
>         self.channels = channels
>         self.patch_size = patch_size
>     def encode(self, x):
>         x = rearrange(x, "b c (l h) -> b (c h) l", h=self.patch_size)
>         return x
>     def decode(self, z):
>         z = rearrange(z, "b (c h) l -> b c (l h)", h=self.patch_size)
>         return z
> ```
> > Have you tried the 50Hz x1 setting? Or, what performance do you expect your method to achieve under this setting?
>
> Yes, we've tried this - although we haven't fully trained a model at this temporal resolution as we were targeting a lower bitrate. The reconstruction performance should be better than the 25Hz model we presented (assuming the bottleneck is the same) as the compression ratio is less.
>
> >I noticed that the number of channels in your discriminator implementation is 256. I'm using the MRD from DAC (link). I tried setting the channels to 256, and the discriminator's parameters are about 100M. Is this correct? How does it compare to the number of parameters in your discriminator?
>
> We're using the Encodec discriminator which has less params in general than the DAC discriminator. Total param count is 35.5M.
>
> >Regarding SYSTEMATIC BIAS, I tried it but suspect there may be an issue with my implementation. Here's my code; I'm not sure if I understood your paper correctly:
>
> Your code looks correct, but you'll need to set `normalized = True` in the `torch.stft` call - otherwise it'll produce very large values. This is turned on in Encodec (I'd assume also in DAC) as otherwise you'll get huge activations inside your discriminator.
>
> >About the encoder, I tried using mel + Transformer or STFT + Transformer, but the former resulted in worse performance, and the latter's performance was similar to DAC's encoder (about 40M parameters). I'm not sure if your method has experimented with different encoders. If you still use the CNN-based DAC encoder, is there a significant difference in your method? My thinking about the encoder is that since the encoder's features have to go through VQ, it means that no matter how powerful your encoder is, after VQ, it can only retain key information. Does this mean that for the encoder, maybe we don't need a very powerful model—perhaps DAC's encoder is sufficient?
>
> Our feeling is that a powerful encoder actually helps the model effectively communicate information through the very restrictive FSQ bottleneck. We did experiment with smaller encoders, but we found that this makes the decoder act more like a generative model - we could still get plausible speech output but the alignment with the input was less precise. Investigating this relationship in more detail (and if it holds with VQ instead of FSQ) could be a very interesting line of future research.
>
> Thanks again for your comments! Happy to discuss more if you have further questions.

---

### Meta-Review · Area_Chair_xrrA · 2024-12-22

**Metareview:**

**Paper Summary:**

This paper describes low-bitrate neural discrete audio codec. This is achieved by scaling up an encoder-decoder Transformer architecture using finite scalar quantization (Mentzer et al., ICLR 2024) in contrast to the more traditional a VQ-VAE discretization. Experimental results are strong for both quantitative metrics and human evaluation.

**Strengths:**

There is general consensus that the empirical analysis is strong, demonstrating a new state-of-the-art in low bit rate speech compression.

All reviewers praised the paper's insights. There is also mostly positive praise for the paper's clarity and organization.

The authors are committed to releasing the artifacts of this work, which will be of great value to the community.

**Weaknesses:**

As Reviewers 1mKc and Baq6 point out, achieving better results by scaling up neural networks is no longer a surprising result for the community. On the other hand, this work identifies and documents many valuable insights into the architectural decisions required to effectively scale neural audio codecs.

**Additional Comments On Reviewer Discussion:**

Authors addressed most of the concerns raised by reviewers in the discussion period, and have revised the paper accordingly to reflect these highly productive discussions. The paper has improved considerably over the course of the reviewing cycle.

---

### Decision · Program_Chairs · 2025-01-22

Accept (Poster)